# Construction and Evaluation of a Safe Community Evaluation Index System—A Study of Urban China

**DOI:** 10.3390/ijerph191710607

**Published:** 2022-08-25

**Authors:** Chao Feng, Jingjie Wu, Juan Du

**Affiliations:** 1School of Public Administration, Northwest University, Xi’an 710127, China; 2School of Economics, Northwest University of Political Science and Law, Xi’an 710063, China; 3School of Management, Northwestern Polytechnical University, Xi’an 710072, China

**Keywords:** urban public safety, safe community, sustainable development, disaster prevention and mitigation, risk factors

## Abstract

A community is the basic unit of a city. Scientific and effective evaluations of the construction effect of safe communities can improve the construction capacity of community disaster prevention and mitigation; it is also the basis for improving urban public safety and realizing stable and sustainable urban operation. First, following the development framework of a safe community and taking two typical communities in Xi’an, China, as examples, based on the literature and expert opinions, the initial indicators of a safe community are determined. Second, based on existing data, the literature and expert opinions, a questionnaire is designed, and the reliability and validity of the questionnaire are tested by exploratory factor analysis. Third, the indicators for evaluating the construction ability of a safe community are selected. Finally, an evaluation model of the construction ability of safe communities is constructed by using the comprehensive weighting technique for order of preference by similarity to the ideal solution (TOPSIS), which is applied to the actual evaluation of eighteen representative communities in Xi’an. The main findings are as follows. (1) The sense of community security is the collective consciousness of community residents. It includes not only the security and feelings of community residents themselves, but also the cognition of the impact of social policies at the macro and micro-levels on community residents, their families, and even the whole community. (2) From the three levels of consciousness, technology, and policy as the starting points for the construction of the theoretical model of a safe community, organizational resilience, accessibility resilience, social environmental resilience, and capital resilience are found to be the main influencing factors in the construction of a safe community. (3) Using questionnaires and expert interviews to preliminarily screen evaluation indicators and using the comprehensive weighting TOPSIS method to build an evaluation model can effectively avoid the defects of traditional empirical research on the validity and reliability of methods. (4) The ranking of the eighteen representative communities in the empirical analysis is basically consistent with the selection results of the national comprehensive disaster reduction demonstration community, which indicates the effectiveness and accuracy of the indicators and algorithms.

## 1. Introduction

With the improvement in China’s urbanization rate, cities continue to expand, and a large number of people continue to cluster together. Subsequently, urban security issues have become prominent. As a complex system, cities are inevitably impacted by a variety of risks. When a variety of events are superimposed, the outcome of Chinese cities will have an enormous impact on the normal operation of cities as a whole if they do not have enough capacity to withstand an impact; furthermore, they will not be able to save themselves and recover from the impact. A community is the basic unit of a city, and disasters or risks are often perceived first at the community level. Therefore, studying the mechanism of community operation under the external disturbance of various events and exploring the influencing factors for building a safe community are of great significance for more scientific and effective intervention and control of crises, as well as for ensuring the safe, stable, and sustainable operation of the community.

Foreign countries carried out the construction of safe communities relatively early. For example, Berkeley in the San Francisco Bay area of California drafted a community safety strategy to improve its economic and social sustainability in a high-risk environment. To encourage homeowners to take action for earthquake safety, the program approved a real estate transfer tax rebate plan and a license fee rebate plan and provided loans and a free home maintenance program for low-income elderly and disabled people. Berkeley has the highest building reinforcement rate in the San Francisco Bay area, is known as the “impact project” community, and uses the seed funds of this project to establish a partnership between safe communities and regions [1]. Singh et al. [2] proposes a framework aimed at quantifying the disaster resilience of urban systems while ensuring an adequate level of sustainability, all according to a social and human-centric perspective, urban networks are modelled as hybrid social–physical networks (HSPNs) by merging both physical and social components, and engineering measures are performed on HSPNs as a measure of urban efficiency within a multiscale approach. Some scholars indicate that social indicators are identified to characterize quality of life in the aftermath of a catastrophic event [3,4]. Both efficiency and quality of life indicators are evaluated using a time–discrete approach before and after an extreme event occurs and during the recovery phase to measure inhabitant happiness and environmental sustainability [5,6]. Some scholars have also studied the resilience of communities to flood resistance [7,8,9], community resilience to earthquakes [10,11], community resilience to windstorms [12,13], and the methodology for evaluating community resilience [14,15,16].

Domestic practical experience in the construction of safe communities is mainly used for reference, and the construction mode of safe communities in line with China’s national conditions is studied mainly by referring to foreign construction experience. For example, Wang et al. [17] summarized the construction work of safe communities in Great Britain, the United States, and Japan and proposed that the construction of safe communities in China should have “rebound” and “reorganization” abilities, including the absorption of external disturbances. In practice, the coordination between self-organization and other organizations, the ability to quickly restore normality, and the ability to cope with the combination of software and hardware should focus on top-down, bottom-up, and comprehensive promotion. The construction of resilient cities in China has just begun, and the construction of resilient communities has not attracted enough attention. There is room for improvement in many aspects, such as the establishment of a resilient community agenda, community public spaces, and community governance [18]. Peng et al. [19] summarized the construction of safe communities abroad from three aspects, i.e., community development, planning, and management, and they discussed the promotion strategies of safe communities in China from three dimensions, i.e., theoretical research, policy guidance, and practice. Huang [20], based on a case study of an old settlement of the Kucapungane (Rukai) people in Taiwan who experienced a forced relocation driven by the 2009 Typhoon Morakot, argued that heritage preservation serves as a link connecting the past and the future, through which communities have a better chance to orient themselves in navigating displacement and participating in post-disaster recovery.

In terms of the theoretical model of building a safe community, the most representative models are those proposed by Norris et al. in 2008 and the well-being, identity, services, and capitals (WISC) model proposed by Scott B. Miles in 2015. Norris et al. [21] believe that a safe community (community disaster resilience) is a process that combines an adaptability network (resources with dynamic attributes) with adaptation after a disturbance or adversity. It is an elastic theory, including the understanding of stress, adaptation, health, and resource dynamics, which can be evaluated by the health of the community population (including mental health, behavioral health, quality of life, etc.). The WISC model is named after four structures, i.e., well-being, identity, services, and capitals, which represent happiness, identity, services, and capitals, respectively [22]. These four structures are defined by 29 variables. The WISC model proposes how to evaluate safe communities, how to collect data, etc. With the continuous improvement in the academic identity of a safe community, the construction, evaluation, and measurement of the index system for safe communities has become the main trend of current research on community resilience.

In terms of the construction of a safe community index system, foreign researchers often build evaluation indicators based on different purposes, such as community capital, social resilience, economic resilience, ecological resilience, disaster prevention facilities resilience, and housing facility resilience, and they often propose a series of evaluation indicators involving society [23], the economy [24], nature, spatial management [25], etc. Based on the current community spatial structure [26], system policy [27], management mode, and other comprehensive indicators [28,29,30,31], the index system is highly scientific and applicable. Not only are Chinese communities quite different from foreign communities in terms of their management mode, spatial structure, and policies, but residents’ living habits are also quite different. It is not feasible to directly learn from the foreign index system of safe communities, which requires Chinese researchers to make corresponding changes based on China’s national conditions. Among foreign index systems, the more representative are the constituent elements and index system of community resilience [32], the evaluation index system of community flexibility from the perspective of emergency management [33], the disaster resilience index of communities from the perspective of fire safety [34], and the resilience measurement of urban communities under the background of climate change [35].

In terms of safe community index system measurement and evaluation, based on the different attributes of research methods, the measurement methods for community resilience in foreign countries are mainly qualitative and quantitative [36,37,38,39]. Among them, questionnaires, the analytic hierarchy process, the comprehensive index method, and social network analysis are commonly used as quantitative methods. In China, the measurement of safe communities is often regarded as the core of research and has attracted increasing attention from researchers. On the basis of summarizing the research of foreign mainstream journals and community resilience measurement, Chinese scholars explore the applicability under China’s national conditions [40,41,42,43,44,45]. There are also some scholars who learn from the results of foreign research and build on them to meet the national conditions of China. For example, Hu et al. [46], analyzed the application of measurement tools in the Chinese context by comparing them. Yang et al. [47] analyzed the social ecosystem of the Qinling Mountains from the perspective of community resilience. Zheng et al. [48] performed a comparative analysis of the safety toughness of communities with high-frequency sudden disasters based on the technique for order of preference by similarity to the ideal solution (TOPSIS). From a qualitative perspective, Shi et al. [49], divided the evaluation indicators of community disaster prevention resilience into five levels, i.e., organizational resilience, social resilience, economic resilience, social capital, and facility resilience, and they elaborated on the selection principles, basis, and specific index content of the evaluation indicators. Other scholars have studied toughness evaluation and optimization from the perspective of risk [50,51].

With the continuous advancement of different theories, measurement methods, and indicators, there are certain differences between the corresponding research methods. At present, most community resilience index systems constructed by current research have certain defects in the empirical validity and reliability of the method. There are mainly three aspects.

First, most of the safe community indicator systems are constructed by means of the literature, expert interviews, and questionnaires, without examining the corrected item total correlation (CITC) of each indicator, especially verifying that the indicator is deleted. If the indicator is deleted, α coefficient is significantly increased, this indicates that the indicator is not very representative of the construction of safe communities and should thus be deleted. This study verifies the CITC of 35 initial tertiary indicators screened through the literature and expert interviews, as well as the 35 α coefficients after deleting the 35 initial tertiary indicators. If the indicator with a CITC less than 0.5 is deleted, the α coefficient will be greatly improved. This indicates that the indicator with a CITC of less than 0.5 cannot represent the safe community well and should thus be deleted, thereby enhancing the accuracy of the index construction. 

Second, in terms of weight setting, the analytic hierarchy process (AHP) is mostly used in safety community studies. Although this method is simple and practical, it relies heavily on expert scoring and is highly subjective. In particular, in situations consisting of too many indicators, large data statistics, and uncertain weights, the analysis matrix is too large to be solved. Through the comprehensive weighting TOPSIS method, the subjective and objective weights are comprehensively weighted to weaken the influence of subjective factors on the results to enhance the reliability and accuracy of the study.

Finally, research on safe community indicators in China has just started. The only research focuses on the perspective of management, thus, focus on related issues, such as how to build the measurement and evaluation system of community resilience, is rare. In view of the fact that the current index system for evaluating safe communities in China mostly considers the hardware infrastructure for disaster prevention and mitigation, and little or no consideration is given to the residents’ safety feelings and the impact of relevant policies, such as employment, medical care, further education, and old-age care on the residents, the analysis is not systematic enough, and the impact of different factors on safety community construction is not given much attention. Therefore, the index system constructed by the questionnaire, field investigation, and interview methods considers the relevant safety feelings of residents, thus enhancing the practicability of the safety community evaluation index system and providing a scientific basis for decision-making for managers and enriching management countermeasures.

Accordingly, this research follows the framework of the resilient development of safe communities, takes typical communities in Xi’an, China, as actual cases, uses questionnaires, expert interviews, factor analysis, and the comprehensive weighting TOPSIS method to screen evaluation indicators and build evaluation models to find the main influencing factors in building a safe community, avoids the defects in the validity and reliability of traditional empirical research, and improves the ability of cities to respond to, deal with, and recover from emergencies.

## 2. Dimension Definition and Evaluation Index Sifting

### 2.1. Dimension Definition

In 2005, Professor Gao Feng and Professor Zhu Yuguo argued in their research on the sense of community security of Shanghai citizens that the “sense of community security” is community residents’ feeling about public security in the areas where they live, their subjective cognition of the dynamic balance between the destructive power and control of social security, and a kind of group consciousness [34,35]. 

We conducted on-the-spot investigation and resident interviews within the X and Y communities in Xi’an, Shaanxi Province. This research of the X and Y communities was funded by the China Postdoctoral Science Foundation (grant number: 2020M673462). We found that the sense of community security can have a broader meaning: community residents not only pay attention to community security, but also pay more attention to the content of social policies, such as the education and employment of children around the community, the charging of medical thresholds near the community, community building facilities, hidden dangers and environmental safety, community pollution issues and other policies closely related to the production and life of community residents, the city’s medical insurance and pension policies, the city’s unemployment and employment policies, the city’s environmental pollution, and other livelihood engineering issues. Macro and micro-level policies can attract more attention from community residents than community security issues.

“Resilience community” is the latest urban governance concept. Resilience is not a single concept but rather a generalization of a system framework. The purpose of community construction under the framework of this system is to improve the community’s ability to respond positively to crisis, adaptability, and sustainable development. Through on-the-spot investigation and resident interviews within the X and Y communities in Xi’an, Shaanxi Province, our community residents are not only concerned about the infrastructure construction of disaster prevention and mitigation, but also macro and micro-level policies which can attract more attention from community residents than community security issues. 

Accordingly, we define the concept of community security as the collective consciousness of community residents, including not only the security and feelings of community residents themselves, but also the cognition of the impact of social policies at the macro and micro-levels on community residents, their families, and even the whole community. According to this, we build the theoretical model of safe community from the three aspects of “consciousness, technology, and policy”. The specific conceptual relationship is shown in Figure 1.

Accordingly, the concept of community security is defined as the collective consciousness of community residents, including not only the security and feelings of community residents themselves, but also the cognition of the impact of social policies at the macro and micro-levels on community residents, their families, and even the whole community. The three levels of consciousness, technology, and policy are the starting points for constructing the theoretical model of community resilience from the perspective of security.

Consciousness refers to the feelings of community residents. It not only includes the basic constituent elements of human feelings, ethics, authority, and the morality of community residents, but also includes the accessibility of residents’ sharing platform built by the community, the degree of interaction in community residents’ interpersonal relationships, the ability of community staff to adjust and resolve community residents’ contradictions, residents’ sense of identity and belonging, etc. By building the emotional resilience of community residents, we can break the limitations of previous studies that did not conduct research from the perspective of the emotional awareness of community residents, and we can pay attention to the real feelings of community residents.

Technology refers to the degree of digitalization, intellectualization, and informatization of community infrastructure. It includes not only the community’s infrastructure and its accessibility, but also the “flat” and “seamless” connection between various departments in the community, the willingness of community residents to participate in public affairs, the use of information technology, etc.

Policy refers to both formal and informal systems and policies. It includes not only the construction of the rule of law and management system in the community, but also the introduction of formal and informal systems and policies by relevant social departments. For example, through on-the-spot investigation and resident interviews within the X and Y communities in Xi’an, Shaanxi Province, we found that previous research found that community residents are concerned about policies related to children’s education and employment, the city’s medical insurance and pension policies, and the city’s unemployment and employment policies.

### 2.2. Initial Evaluation Index Sifting

Based on the analysis of the previous literature, the selection of initial indicators of safe communities follows the five principles of scientificity, representativeness, operability, applicability, and comprehensiveness. The process of selecting the initial indicators of safe communities is divided into the following four steps. 

**First,** selection was based on the induction and analysis of the relevant literature and practical cases in China and other countries. Marked tertiary indicators in Table 2 are from the existing literature and expert interviews, and the other part is the discussion and research of our team.

**Second,** cluster analysis on questionnaire survey data and network big data collected in the early stage was carried out to screen the influencing factors that affect the safety feelings of community residents. The data source includes two parts: one part is from the survey data including from the questionnaire, expert interviews, and historical data collation; the other part comes from big data on the internet including “local treasure” in Xi’an city, “community owners forum”, and “WeChat apps group” in X community and Y community, as well as other relevant web pages, forums, and WeChat apps. It mainly completes the following three aspects of work. (1) Supplement the lack of questionnaire and expert interview data, so as to obtain accurate and real-time data and expand the data set. (2) Discover the change and evolution of the residents’ mentality in the face of specific disasters and provide scientific data support for the final policy recommendations. (3) Provide help for building community resilience indicators, and obtain relevant data information such as user publishing time, emergency keywords, emotion words, and place words from the network. The key data summary obtained from these relevant web pages and forums is shown in Table 1.

The data collected by the web crawler program written in Python is preprocessed. The preprocessing work is mainly composed of two parts. (1) The text is de-duplicated, the mechanical compression is de-worded, and the short sentence is deleted to remove the unhelpful sentences and words. (2) The data is cleaned, integrated, transformed, and regulated, and then the simulation analysis data required by this topic is constructed. The specific steps are shown in Figure 2.

**Third,** the project team held many internal meetings to analyze, screen, supplement, revise, and update the influencing factors and indicators obtained in the steps above. Unmarked tertiary indicators in Table 2 are from the discussion and research of our team.

**Fourth,** the opinions and suggestions of relevant experts in academia and industry were conducted, and the index system was revised and improved. The work of this part can be seen in my student’s dissertation (Master’s thesis) [52].

The initial indicators are shown in Table 2.

### 2.3. Determination of Evaluation Index

#### 2.3.1. Preparation and Distribution of Questionnaires

To verify the reliability and accuracy of the 5 primary indicators, 13 secondary indicators, and 35 tertiary indicators that were screened out, we designed a questionnaire that contained the initial 35 indicators, and the questionnaire was divided into two parts. The first part of the survey consisted of questions that focus on the basic personal information of community residents and community managers, including their gender, age, educational level, type of housing, building form of housing in the community, building age, and time living in the community. The second part uses a Likert scale to measure satisfaction, with answers ranging from very satisfied (5 points), satisfied (4 points), indifferent (3 points), dissatisfied (2 points), and very dissatisfied (1 point).

The questionnaire was distributed to the community residents and community managers located in communities X and Y in Xi’an, the community staff members in charge of the emergency departments, and the professional scholars working in this field at universities located in Xi’an. For community residents and community managers, 340 questionnaires were distributed within a week by using the Questionnaire Star and WeChat apps. For the community staff in charge of the emergency departments and the experts and scholars working in this field at universities located in Xi’an, a total of 20 questionnaires were distributed within a week by using the Questionnaire Star and WeChat apps. For elderly individuals over 60 years old in the community, a paper version of the questionnaire was used, 40 questionnaires were distributed within a week, and random sampling was used to find participants in the community.

The X community was established in 2000.The total area of the jurisdiction is 0.6 square kilometers, with 4416 permanent residents and a population of 11,700. There are 21 courtyards and 5 units. There are 15 community work service personnel, 8 public welfare posts, 1 Party branch, 3 Party branch members, 169 Party members, and 7 on-the-job Party members; there are 8 members of the neighborhood committee and 3 full-time community members (cross employment). The office area of the community is 336 square meters. There is a one-stop service hall, a Party building activity room, a reading room, and an activity room for middle-aged and elderly individuals in the community, which can provide various convenience services for community residents, such as Party member services, labor security, and floating population management. The X community adheres to the principle of harmonious community construction, takes serving community residents as the core, and takes sustainable development of residents’ autonomy and strengthening neighborhood harmony as the goal. It integrates resources and establishes distance education broadcasting points, popular science universities, population-based schools, staff bookstores, and other characteristic lectures to disseminate knowledge and improve the quality of the masses. The community hosts parties on cool evenings, community sports games, Spring Festival couplets, Chongyang saozi noodles events, the Winter Solstice Dumpling Banquet, and other activities to tighten the relationship between the Party and the masses and between the cadres and the masses. To improve residents’ level of happiness, grid management and the dean system are implemented, and work efficiency is advocated. This approach presents a new outlook of “residents’ autonomy, orderly management, perfect service, good public order, beautiful environment, civilization, and harmony”. The X community has won the honorary titles of a national disabled community rehabilitation demonstration site, a Shaanxi four-star community party organization, a Shaanxi safe family demonstration community and so on.

The Y community was rated as an affordable housing project by the Xi’an municipal government in 2004. It covers an area of 0.11 square kilometers. The total construction area of the community is more than 200,000 square meters. There are 2719 permanent residents and a population of 9327 people. There are kindergartens, open clubs, supermarkets, clinics and gyms in the community, of which the kindergartens cover an area of 1500 square meters and the open clubs cover an area of 1900 square meters. There are seven educational institutions positioned around the community. The water supply and power supply are under the centralized management of the municipal government, and all residential heating pipes are designed and installed in a household-by-household manner. This is not only conducive to household measurement after the public network transformation but can also regulate the indoor heating temperature. There are 10 community work and service personnel in Y community, and a Party branch organization has been established. The community has an office area of 253 square meters. The office area has a variety of functions, such as a one-stop service hall, a Party building room, and an activity room for elderly individuals. This area helps not only to serve community residents, but also to maintain the community environment; it also provides more labor security, civil affairs assistance, family planning, and other convenient services for community residents. As the community is located close to Cultural Park, it has more coverage of green plants, which makes the community appear more humanized.

A total of 400 questionnaires were distributed. After eliminating the invalid questionnaires, 374 valid questionnaires were finally obtained, with an effective recovery rate of 93.5%. Descriptive statistical analysis of the samples is shown in Table 3.

#### 2.3.2. Questionnaire Results and Analysis

Statistical Product Service Solutions (SPSS) software (IBM, New York, NY, USA) was used to test the reliability and validity of the samples. The specific results are shown in Table 2. The Cronbach’s alpha coefficient of each dimension of safe communities is higher than 0.8, indicating that the reliability of the questionnaire is good. The Kaiser–Meyer–Olkin (KMO) value is 0.874, and the significance of Bartlett’s test of sphericity is 0.000, indicating that the questionnaire has good validity. 

The corrected item-total correlation (CITC) values of the corrected items are greater than 0.5, indicating that there is a good correlation between the items and a good level of reliability. However, the CITC of the frequency of community disaster prevention and reduction testing and early warning facilities (*C*_5_), community organization self-rescue (*C*_6_), social assistance (*C*_23_), the per capita effective refuge area (*C*_28_), and community day nursery care (*C*_35_) are all less than 0.5. If these five items are deleted, then the reliability coefficient will be greatly improved. It shows that the five tertiary indicators cannot well represent the relevant influencing factors of safe communities. Therefore, these five items are deleted, and the specific results are shown in Table 4.

#### 2.3.3. Factor Extraction

Because exploratory factor analysis is used, the analysis using principal component analysis and Caesar’s normalized maximum variance method converges after six iterations. In the factor extraction, principal component analysis in the exploratory factor analysis method is used to extract the common factors, and the dimensionality of the primary indicators is reduced. The common factors are extracted through varimax rotation, and the rotated factors are analyzed to obtain the eigenvalues of the correlation matrix, the variance contribution rate, and other data. The eigenvalues of the four common factors are greater than 1.000. According to the principle of selecting common factors, four common factors are selected. The variance explained rates of the four factors after rotation are 24.898%, 19.689%, 9.969%, and 9.732%, respectively. The cumulative variance explained rate after rotation is 64.289%, which is greater than 50%, meaning that most of the information of the data as a whole can be summarized and basically explained. See Table 5 for the characteristic values and explained rates.

The loading coefficient of the analysis factor is basically in line with the expectation. Therefore, based on the common characteristics of the variables with higher loadings among the common factors, four common factors are finally extracted as first-class indicators, named organizational toughness (*F*_1_), accessibility toughness (*F*_2_), the social environment (*F*_3_), and capital willingness (*F*_4_).

We initially screened 5 primary indicators (*A*_1_, *A*_2_, *A*_3_, *A*_4_, *A*_5_), 13 secondary indicators, and 35 tertiary indicators. After the verification of a series of methods such as questionnaires, 30 tertiary indicators were selected. After factor analysis, only four factors (*F*_1_, *F*_2_, *F*_3_, *F*_4_) were basically in line with the expectation. The factors (*F*_1_, *F*_2_, *F*_3_, *F*_4_) here are the same as the primary indicators (*A*_1_, *A*_2_, *A*_3_, *A*_4_). We use factors (*F*_1_, *F*_2_, *F*_3_, *F*_4_) to represent the common factors.

Because *C*_28_, *C*_29_, *C*_30_ tertiary indicators are important for research, especially for elderly care and disaster prevention and mitigation, *C*_28_, *C*_29_, *C*_30_ tertiary indicators cannot be classified into other primary indicators. The primary indicator *A*_5_ was retained.

#### 2.3.4. Revised Indicators

In the revised indicators, individual items are deleted. Finally, 5 primary indicators, 13 secondary indicators, and 30 tertiary indicators are screened out and classified. After testing, the selected indicators are reasonable and effective. The safe community evaluation index system is shown in Figure 3. Figure 3 is a diagram of the final indicators screened in Table 1 through questionnaire analysis and verification.

## 3. Construction of a Safe Community Assessment Model Based on the Comprehensive Weighting TOPSIS Method

The abovementioned screened-out indicators are used to build a safe community evaluation index system. The index elements not only take into account the safety feelings of community residents, but also address all types of natural disasters, which is convenient for resilience evaluation. 

### 3.1. Establishment of the Decision Matrix

Suppose that there are *k* communities participating in the evaluation and *n* evaluation indicators, forming an evaluation matrix with *n* rows and *k* columns:(1)$$X={\left(x_{ij}\right)}_{k\times n}=\left[\begin{array}{ccccc} x_{11} & x_{12} & x_{13} & \ldots & x_{1n}\\ x_{21} & x_{22} & x_{23} & \ldots & x_{2n}\\ x_{31} & x_{32} & x_{33} & \ldots & x_{3n}\\ \ldots & \ldots & \ldots & \ldots & \ldots \\ x_{k1} & x_{k2} & x_{k3} & \ldots & x_{kn} \end{array}\right]$$

*x_ij_* is the *j*-th index of the *i*-th community, where *i* = 1, 2, 3, …, *k*; *j* = 1, 2, 3, …, *n*.

### 3.2. Index Normalization and Standardization 

Each indicator is divided into maximum, minimum, intermediate, and interval indicators according to its own attributes, and the evaluation matrix needs to be normalized and standardized. All tertiary indicators in the evaluation matrix *X* are converted into normalized indicators, that is, unified into maximum indicators, and a normalized matrix is obtained. The following describes the forward processing of different types of indicators.

(1)For maximum tertiary indicators, we do not need to perform any forward processing.(2)For minimum tertiary indicators (the smaller the indicator value is, the better), among the 30 tertiary indicators screened in this study, *C*_17_ (population density) and *C*_19_ (the proportion of special populations) are minimum tertiary indicators. The tertiary indicator *C*_17_ (population density) is obtained from the statistical yearbook of each district. We believe that the smaller the population density is, the richer the per capita public resources are, which is relatively good. Of course, the population density cannot be infinitely small. The tertiary indicator *C*_19_ (the proportion of special populations) is obtained from the statistical yearbooks of various districts. The special population refers to disabled individuals, elderly individuals, and other related groups. We believe that the smaller the proportion of the special population, the stronger the community’s ability to resist disasters. We convert the minimum indicators to the maximum indicators:
(2)$$\tilde{x_{i}}=max\left\{x_{i}\right\}-x_{ij}$${*x_i_*} is a group of minimum index series, $\tilde{x_{i}}$ is the converted indicator.(3)For intermediate tertiary indicators (intermediate tertiary indicators are indicators whose values should not be too large or too small, and the closer to a certain value, the better), among the 30 tertiary indicators screened in this study, there are no intermediate indicators. However, for the sake of the integrity of the study, if there are intermediate indicators, the intermediate indicators are converted into maximum indicators:
(3)$$M=max\left\{\left|x_{i}-x_{best}\right|\right\},\;\tilde{x_{i}}=1-\frac{\left|x_{i}-x_{best}\right|}{M}$${*x_i_*} is a set of intermediate index series, and the best value is *x_best_*, $\tilde{x_{i}}$ is the converted indicator.(4)For interval tertiary indicators (interval tertiary indicators are the best indicators that their values fall within a certain interval), among the 30 tertiary indicators screened in this study, *C*_13_ (refuge), *C*_22_ (building density), and *C*_23_ (the greening rate) are interval tertiary indicators. *C*_13_ (refuge) is obtained from the statistical yearbook of each district. Refuge refers to the effective per capita refuge area. According to the relevant regulations of Xi’an, the effective per capita refuge area in Xi’an is 1.5–3 m^2^/person, which is an interval type three-level indicator. *C*_22_ (building density) is obtained from the statistical yearbook of each district. Building density refers to the per capita effective refuge area. According to the relevant regulations of Xi’an, the building density of the Xi’an community is 20–39%, which means that it can maximize the use of space and public resources. *C*_23_ (the greening rate) is obtained from the statistical yearbook of each district. The greening rate is an indicator used to measure the greening degree of community roads. According to the relevant regulations of Xi’an city construction, the greening rate of the community is 20–30%, which makes the residents feel better. Interval tertiary indicators need to be positively processed; that is, interval tertiary indicators are converted into maximum tertiary indicators:
(4)$$M=max\left\{a-min\left\{x_{i}\right\},max\left\{x_{i}\right\}-b\right\},\;\tilde{x_{i}}=\{\begin{array}{cc} 1-\frac{a-x_{i}}{M}, & x_{i}<a\\ 1, & a\leq x_{i}\leq b\\ 1-\frac{x_{i}-b}{M}, & x_{i}>b \end{array}$${*x_i_*} is a set of intermediate index series, and the best value is [*a*, *b*].

In addition to the abovementioned tertiary indicators, i.e., *C*_13_, *C*_17_, *C*_19_, *C*_22_, and *C*_23_, the other 25 tertiary indicators are maximum tertiary indicators and do not need any forward processing. To eliminate the influence of different dimensions, the normalized index matrix is standardized, and the normalized evaluation matrix is recorded as *Z*.
(5)$$Z={\left(z_{ij}\right)}_{k\times n}=\left[\begin{array}{ccccc} z_{11} & z_{12} & z_{13} & \ldots & z_{1n}\\ z_{21} & z_{22} & z_{23} & \ldots & z_{2n}\\ z_{31} & z_{32} & z_{33} & \ldots & z_{3n}\\ \ldots & \ldots & \ldots & \ldots & \ldots \\ z_{k1} & z_{k2} & z_{k3} & \ldots & z_{kn} \end{array}\right]$$
(6)$$z_{ij}=x_{ij}/\sqrt{\sum_{i=1}^{n}x_{ij}^{2}}$$

### 3.3. Weight Calculation

We determine the index weight *W*. To weaken the influence of subjective factors on the results, the subjective and objective combination weighting method is used to weight the indicators. The subjective part adopts the analytic hierarchy process (AHP), the objective part adopts the variation coefficient method, and finally the subjective and objective preference coefficient is introduced to obtain the comprehensive weight.
(7)$$W=\beta W^{*}+\left(1-\beta \right)W^{\prime }={\left[w_{1},w_{2},w_{3},\ldots ,w_{n},\right]}^{T}$$

*W** is the weight obtained by AHP method, *W*’ is the variation weight obtained by the variation coefficient method, and *β* is the subjective and objective preference coefficient, *β* = 0.5. The variation weight *W*’ can be obtained by the variation coefficient method, and the calculation process is as follows:(8)$$V_{j}=\frac{S_{j}}{\tilde{x_{j}}}$$

*V_j_* is the coefficient of variation of matrix *Z*, *S_j_* and $\tilde{x_{j}}$ are the standard deviation and mean value of matrix *Z*. Normalize *V_j_*:(9)$$w_{j}^{\prime }=\frac{V_{j}}{\sum_{j=1}^{n}V_{j}}$$

*w_j_* is the variation weight of the *j*-th index.
(10)$$W^{\prime }={\left[w_{1}^{\prime },w_{2}^{\prime },w_{3}^{\prime },\ldots ,w_{j}^{\prime }\right]}^{T}$$

### 3.4. Evaluation Ranking

We calculate the weighted distance from each evaluation object to the ideal solution $D_{i}^{+}$ and the negative ideal solution $D_{i}^{-}$:(11)$$\begin{array}{l} Z^{+}=(Z_{1}^{+},Z_{2}^{+},Z_{3}^{+},\ldots ,Z_{m}^{+})\\ =(max\left\{z_{11},z_{21},z_{31},\ldots ,z_{n1}\right\},max\{z_{12},z_{22},z_{32},\ldots ,z_{n2}\},max\{z_{13},z_{23},z_{33},\ldots ,z_{n3}\},\ldots ,max\left\{z_{1m},z_{2m},z_{3m},\ldots ,z_{nm}\right\}) \end{array}$$
(12)$$\begin{array}{l} Z^{-}=(Z_{1}^{-},Z_{2}^{-},Z_{3}^{-},\ldots ,Z_{m}^{-})\\ =(min\left\{z_{11},z_{21},z_{31},\ldots ,z_{n1}\right\},min\left\{z_{12},z_{22},z_{32},\ldots ,z_{n2}\right\},min\left\{z_{13},z_{23},z_{33},\ldots ,z_{n3}\right\},\ldots ,min\left\{z_{1m},z_{2m},z_{3m},\ldots ,z_{nm}\right\}) \end{array}$$

The distance between the *i*-th (*i* = 1, 2, 3, …, *n*) evaluation object and the maximum value is:(13)$$D_{i}^{+}=\sqrt{\sum_{j=1}^{m}w_{j}{\left(Z_{j}^{+}-z_{ij}\right)}^{2}}$$

The distance between the *i*-th (*i* = 1, 2, 3, …, *n*) evaluation object and the minimum value is:(14)$$D_{i}^{-}=\sqrt{\sum_{j=1}^{m}w_{j}{\left(Z_{j}^{-}-z_{ij}\right)}^{2}}$$

The *i*-th (*i* = 1, 2, 3, …, *n*) non normalized score of the evaluation object:(15)$$S_{i}=\frac{D_{i}^{-}}{D_{i}^{+}D_{i}^{-}}$$

Obviously $0\leq S_{i}\leq 1$, and $S_{i}$ the larger $D_{i}^{+}$ is the smaller, the closer to the maximum, normalize $\tilde{S_{i}}$: (16)$$\tilde{S_{i}}=S_{i}/\sum_{i=1}^{n}S_{i},$$

Ranking the results and ranking the values from large to small, we obtain the order of the safety evaluation of each community. The algorithm flow chart is shown in Figure 4.

## 4. Practical Application

This study chose Xi’an city as the research object. 

First, this research was funded by the China Postdoctoral Science Foundation (grant number: 2020M673462) and the Social Science Foundation of Shaanxi Province (grant number: 2021R028). The research objects of these two funds are communities in Xi’an.

Second, we participated in the whole process of the “Shaanxi 2021 comprehensive disaster reduction demonstration community evaluation”. During the evaluation, we found that most communities in Xi’an have built emergency shelters, organized emergency drills, investigated potential disaster risks, and built micro fire stations. Through the establishment of comprehensive disaster reduction demonstration communities, the community’s emergency response ability to disasters has been further improved, and the residents’ awareness of disaster prevention and reduction has also been strengthened. However, the existing indicator system of comprehensive disaster reduction demonstration community construction can no longer meet the current needs; it is urgent to increase the indicators of relevant policies that consider and guarantee the residents’ safety feelings. In consideration of the practical process of community disaster prevention and mitigation assessment we participated in, and the problems faced by Xi’an in constructing a safe community, Xi’an city was selected as the research object.

Selection principles and steps of representative communities in Xi’an:(1)According to the “list of comprehensive disaster reduction demonstration communities in Shaanxi Province in 2021” announced by Shaanxi Provincial Emergency Management Department on 3 December 2021, there are 24 communities in Xi’an.(2)According to the recommendation of Shaanxi Province in 2017, “42 communities in Shaanxi Province were selected as national comprehensive disaster reduction demonstration communities”; among them, 6 communities from Xi’an were selected. Since the 24 communities in (1) and the 6 communities in (2) do not all represent the same community, a total of 30 communities were screened.(3)At present, Xi’an is divided into 11 districts and 2 counties. Since the index system of this study involves many factors such as employment, medical care, further education, and old-age care on the residents, according to the distribution of resources such as further education, elderly care, and employment in Xi’an, only the six districts of Xi’an where these resources are concentrated are selected as the community research object (these six districts are Lianhu District, Beilin District, Yanta District, Xincheng District, Weiyang District, and Chang’an District). To ensure the interpretability, fairness, and consistency of the research results, three representative communities were selected for each district. For selection principles see (4)–(5).(4)As most of the existing large-scale communities in Xi’an are “a city within a city”, they face many community safety management problems and are very valuable research objects of safe communities. The population of each district and county is published on the Xi’an Municipal People’s government website. According to the regulations on the size of the community population in China, each district selected a large community with a resident population of more than 30,000.(5)According to the statistical yearbook of Xi’an, approximately 30% of the communities in Xi’an are old. The area of these old communities is relatively small. Block service facilities are generally shared by several communities. The construction quality is low. Most of these community residents are elderly, and their economic capacity is limited. It is a very representative community for a safe community. Therefore, each district selected an old community built prior to the 1980s.(6)Since 2015 and especially in the past three years, to strengthen community capacity building, Xi’an started the construction of “smart communities”. These communities are not limited to “hard” infrastructure, such as living areas, traffic, and the surrounding environment, but “soft” services, such as communication services, security precautions, and external communication, and are gradually becoming the focus of attention. It is also the development direction of the community. Therefore, each district selected a “smart community” that was built after 2015.

According to the above principles and steps, a total of 18 representative communities were selected from 30 communities, and the comprehensive weighting TOPSIS method constructed above was used to evaluate and rank the public security capacity of these 18 communities.

A total of ten experts, including community staff in charge of the emergency department and bureau and professional scholars working in this field in Xi’an colleges and universities, were invited to score the indicators and weights involved in the evaluation of the 30 three-level indicators. The scoring range of each item is 0–100 points, a score of less than 50 points means that the item is very poor, and a score of 50–100 represents different degrees of options from a poor transition to a good transition. After removing the highest and lowest points, the average of the remaining points is taken as the effective score. For example, indicators *C*_2_, *C*_3_, *C*_5_, *C*_7_, *C*_8_, *C*_9_, *C*_10_, *C*_24_, *C*_27_, *C*_28_, and *C*_29_ are needed to be scored by experts; indicators *C*_1_, *C*_4_, *C*_6_, *C*_18_, *C*_20_, *C*_25_, and *C*_26_ are obtained from questionnaires and field research interviews; indicators *C*_11_, *C*_12_, *C*_13_, *C*_14_, *C*_15_, *C*_16_, *C*_17_, *C*_19_, *C*_21_, *C*_22_, *C*_23_, and *C*_30_ are obtained from statistical yearbooks and other relevant data.

The standardized evaluation matrix *Z*.

Forming an evaluation matrix with 18 rows and 30 columns. Equations (1)–(6) are used to standardize the standardization matrix of X. *K* is used for community and the results are shown in Table 6.

2.Weights are calculated.

Equations (7)–(10) are used to calculate weight. The subjective and objective weight calculated by the analytic hierarchy process and coefficient of variation method are shown in Table 7.

3.The results are ranked.

Equations (11)–(16) are used to calculate the results. The weighted distance from each evaluation object to the ideal solution and the negative ideal solution is calculated. The matrix after weighted re-standardization and the optimal solution distance of 18 communities are shown in Table 8.

The safety ability of the eighteen communities is ranked as follows:*K*_12_, *K*_8_, *K*_6_, *K*_9_, *K*_11_, *K*_7_, *K*_10_, *K*_18_, *K*_3_, *K*_14_, *K*_5_, *K*_15_, *K*_16_, *K*_13_, *K*_2_, *K*_1_, *K*_17_, *K*_4_.

We participated in the whole process of the “Shaanxi 2021 comprehensive disaster reduction demonstration community evaluation” by the Shaanxi Provincial Emergency Management Department on 3 December 2021. For the 18 communities screened in this study, the ranking results of the Shaanxi provincial emergency management department are as follows: *K*_12_, *K*_8_, *K*_6_, *K*_7_, *K*_15_, *K*_9_, *K*_14_, *K*_18_, *K*_3_, *K*_10_, *K*_5_, *K*_11_, *K*_16_, *K*_13_, *K*_2_, *K*_1_, *K*_17_, *K*_4_. The comparison results show that the optimal community is basically consistent with the ability results recognized in the industry.

The ranking of *K*_9_, *K*_10_, *K*_11_, and *K*_14_ communities are different. The reason is that the index system we put forward takes into account the feelings related to residents’ safety and the policies of elderly care, medical treatment, further education, and employment that residents are more concerned about. We not only put forward the relevant index system, but also respond with a high weight. At present, the existing safety community evaluation index system does not take these aspects into account, therefore the ranking results of these communities will be different. Security capacity building is basically consistent, which confirms the effectiveness and reliability of the selected community safety ability indicators.

## 5. Conclusions

(1)The resilience of a safe community is the group consciousness of community residents, which includes not only the security and feelings of community residents themselves, but also the cognition of the impact of social policies at the macro and micro-levels on community residents, their families, and even the whole community.(2)The three levels of consciousness, technology, and policy are the starting points for the construction of the theoretical model of safety zone resilience. Using the methods of questionnaire surveys, factor analysis, and expert interviews, from the perspective of residents’ sense of security, the factors affecting the construction of community safety capacity are summarized into four dimensions: organizational resilience (*F*_1_), accessibility resilience (*F*_2_), the social environment (*F*_3_), and capital willingness (*F*_4_). We screened out 11 secondary indicators and 30 tertiary indicators.(3)Using questionnaires and expert interviews to preliminarily screen evaluation indicators and using the comprehensive weighting TOPSIS method to build an evaluation model can effectively avoid the defects of traditional empirical research on the validity and reliability of methods.(4)Through empirical analysis of the representative eighteen communities in Xi’an, we conclude that the order of the construction of the capacity of safe communities is as follows: *K*_12_, *K*_8_, *K*_6_, *K*_9_, *K*_11_, *K*_7_, *K*_10_, *K*_18_, *K*_3_, *K*_14_, *K*_5_, *K*_15_, *K*_16_, *K*_13_, *K*_2_, *K*_1_, *K*_17_, *K*_4_. The results are consistent with the results recognized in the industry, which proves the effectiveness and accuracy of the indicators.

This study has several limitations that should be addressed in future research. On the one hand, the selected areas are only typical communities in Xi’an. Whether they represent the whole country requires further investigation of more regions and provinces. On the other hand, although the sample size meets the minimum sample requirements, obtaining more samples will make the data analysis more accurate. Finally, although several different communities were selected, samples from different communities were not analyzed and compared in detail. Therefore, it is necessary to further explore whether each variable has spatial heterogeneity in future research.

## Figures and Tables

**Figure 1 ijerph-19-10607-f001:**
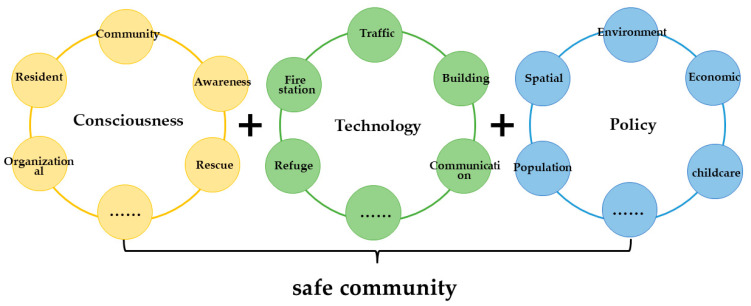
The specific conceptual relationship.

**Figure 2 ijerph-19-10607-f002:**
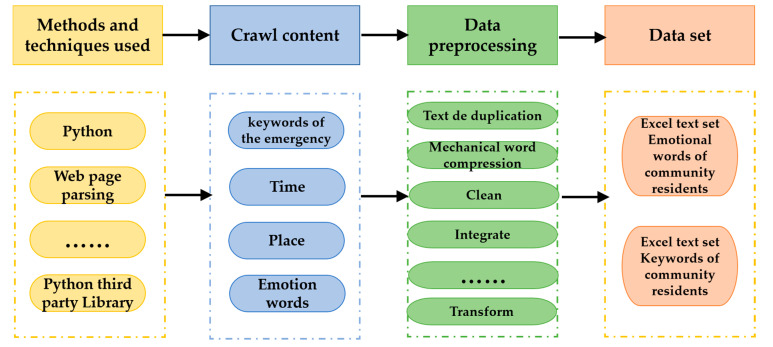
Network big data crawling and processing process.

**Figure 3 ijerph-19-10607-f003:**
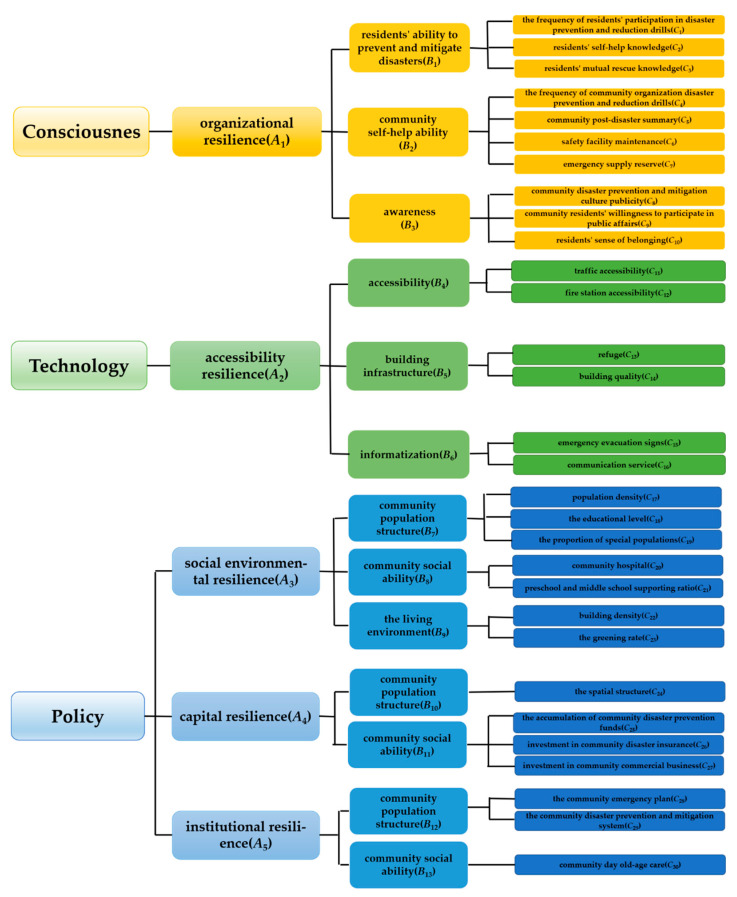
Safe community evaluation index system.

**Figure 4 ijerph-19-10607-f004:**
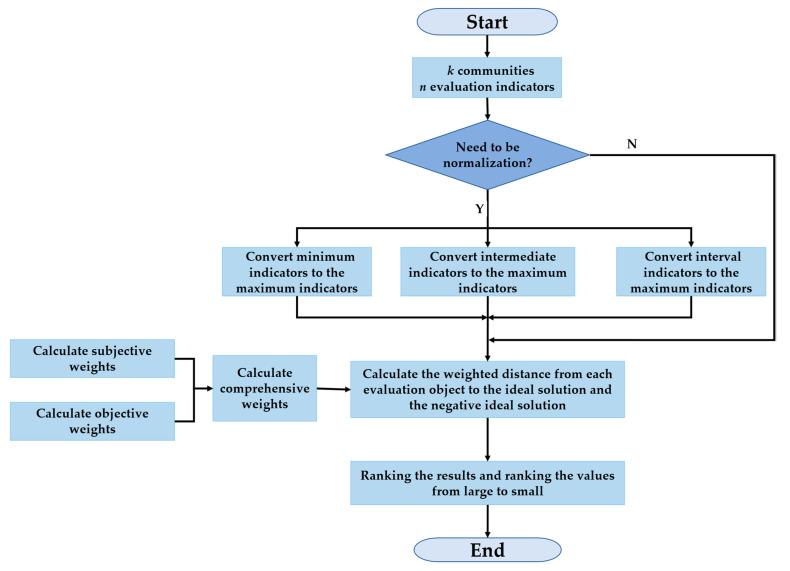
The algorithm flow chart.

**Table 1 ijerph-19-10607-t001:** Key data of relevant web pages and forums.

	Keywords	Time	Place	Emotion Words
**Selection reason**	determine the keywords of the emergency	determine the time node of data crawling	determine the data acquisition site	obtain the change of community residents’ mentality
**Source**	historical data sorting	historical data sorting	interviews with relevant personnel	network

**Table 2 ijerph-19-10607-t002:** Initial indicators of a safe community.

Dimension	Primary Indicators	Secondary Indicators	Tertiary Indicators
Consciousness	organizational resilience (*A*_1_)	residents’ ability to prevent and mitigate disasters (*B*_1_)	the frequency of residents’ participation in disaster prevention and reduction drills (*C*_1_) [49]
			residents’ self-help knowledge (*C*_2_) [49]
			residents’ mutual rescue knowledge (*C*_3_)
		community self-help ability (*B*_2_)	the frequency of community organization disaster prevention and reduction drills (*C*_4_) [48]
			the frequency of community disaster prevention and reduction testing and early warning facilities (*C*_5_)
			community organization self-rescue (*C*_6_)
			community post-disaster summary (*C*_7_)
			safety facility maintenance (*C*_8_) [50]
			emergency supply reserve (*C*_9_)
		awareness (*B*_3_)	community disaster prevention and mitigation culture publicity (*C*_10_) [46]
			community residents’ willingness to participate in public affairs (*C*_11_) [47]
			residents’ sense of belonging (*C*_12_) [21]
Technology	accessibility resilience (*A*_2_)	accessibility (*B*_4_)	traffic accessibility (*C*_13_) [21]
			fire station accessibility (*C*_14_) [21]
		building infrastructure (*B*_5_)	Refuge (*C*_15_) [22]
			building quality (*C*_16_) [22]
		informatization (*B*_6_)	emergency evacuation signs (*C*_17_) [21]
			communication service (*C*_18_) [21]
Policy	social environmental resilience (*A*_3_)	community population structure (*B*_7_)	population density (*C*_19_) [22]
			the educational level (*C*_20_) [21]
			the proportion of special populations (*C*_21_) [22]
		community social ability (*B*_8_)	community hospital (*C*_22_)
			social assistance (*C*_23_) [22]
			preschool and middle school supporting ratio (*C*_24_)
		the living environment (*B*_9_)	building density (*C*_25_) [17]
			the greening rate (*C*_26_) [18]
	capital resilience (*A*_4_)	spatial capital (*B*_10_)	the spatial structure (*C*_27_)
			the per capita effective refuge area (*C*_28_)
		economic capital (*B*_11_)	the accumulation of community disaster prevention funds (*C*_29_) [24]
			investment in community disaster insurance (*C*_30_)
			investment in community commercial business (*C*_31_)
	institutional resilience (*A*_5_)	the disaster prevention and reduction system (*B*_12_) community childcare and pension policy implementation (*B*_13_)	the community emergency plan (*C*_32_)
			the community disaster prevention and mitigation system (*C*_33_)
			community day old-age care (*C*_34_)
			community day nursery care (*C*_35_)

Marked tertiary indicators in Table 2 are from the existing literature and expert interviews, and the other part is the discussion and research of our team.

**Table 3 ijerph-19-10607-t003:** Descriptive statistical analysis of samples.

Basic Information		Mean	Standard	Deviation	Frequency
Gender	male	1.59	0.492	152	40.6%
	female			222	59.4%
Age	Under 18 years old	4.03	0.930	2	0.5%
	19–29 years old			95	25.4%
	30–39 years old			206	55.1%
	40–49 years old			45	12%
	50–59 years old			16	4.3%
	60–69 years old			7	1.9%
	70 years old and above			3	0.8%
Education level	High school and below	2.13	0.581	42	11.2%
	College/undergraduate			242	64.7%
	Postgraduate and above			90	24.1%
House type	Commercial housing	1.56	1.952	310	82.9%
	Housing reform			16	4.3%
	Stock house			5	1.3%
	Fund raising house			11	2.9%
	Housing project			10	2.7%
	Affordable housing			22	5.9%
Building form of community house	Low-rise residence (building height less than 3 floors)	3.53	0.905	62	16.6%
	Multi-storey residence (3–6 floors high)			24	6.4%
	Medium and high-rise residential buildings (7–9 floors high)			268	71.7%
	High-rise residence (the building height is more than 10 floors)			8	2.1%
	other			12	3.2%
Community building year	2000 and before	2.35	0.705	41	11%
	2001–2010			169	45.2%
	2011–2020			155	41.4%
	2021 present			9	2.4%
Residence time	More than 10 years	2.49	0.968	49	13.1%
	5–9 years			170	45.5%
	1–4 years			76	20.3%
	Less than 1 year			79	21.1%

**Table 4 ijerph-19-10607-t004:** Reliability test results of initial indicators.

Tertiary Indicators	CITC	After Deleting Variables α Coefficient
*C* _1_	0.650	0.851
*C* _2_	0.696	0.942
*C* _3_	0.582	0.873
*C* _4_	0.612	0.894
*C* _5_	0.460	0.894
*C* _6_	0.495	0.913
*C* _7_	0.547	0.913
*C* _8_	0.556	0.913
*C* _9_	0.769	0.946
*C* _10_	0.754	0.946
*C* _11_	0.717	0.937
*C* _12_	0.758	0.946
*C* _13_	0.689	0.921
*C* _14_	0.710	0.932
*C* _15_	0.739	0.933
*C* _16_	0.723	0.932
*C* _17_	0.620	0.913
*C* _18_	0.655	0.894
*C* _19_	0.565	0.913
*C* _20_	0.613	0.913
*C* _21_	0.604	0.913
*C* _22_	0.702	0.891
*C* _23_	0.469	0.953
*C* _24_	0.747	0.950
*C* _25_	0.509	0.842
*C* _26_	0.591	0.852
*C* _27_	0.546	0.843
*C* _28_	0.459	0.953
*C* _29_	0.728	0.951
*C* _30_	0.747	0.953
*C* _31_	0.509	0.853
*C* _32_	0.639	0.853
*C* _33_	0.513	0.853
*C* _34_	0.728	0.891
*C* _35_	0.459	0.893

**Table 5 ijerph-19-10607-t005:** Eigenvalues and explained rate.

Common Factors	Before Rotation			After Rotation		
	the Characteristic Values	Initial Eigenvalue Variance %	the Cumulative Variance Explained %	the Characteristic Values	the Variance Explained Rates %	the Cumulative Variance Explained %
*F* _1_	12.099	46.535	46.535	6.474	24.898	24.898
*F* _2_	1.920	7.383	53.918	5.119	19.689	44.588
*F* _3_	1.492	5.739	59.657	2.592	9.969	54.556
*F* _4_	1.204	4.631	64.289	2.530	9.732	64.289

**Table 6 ijerph-19-10607-t006:** The standardization matrix of *Z*.

Communities	Tertiary Indicators									
	*C* _1_	*C* _2_	*C* _3_	*C* _4_	*C* _5_	*C* _6_	*C* _7_	*C* _8_	*C* _9_	*C* _10_
*K* _1_	0.0917	0.2145	0.1941	0.1849	0.2571	0.1302	0.2430	0.2333	0.2397	0.2452
*K* _2_	0.1833	0.1532	0.1509	0.1849	0.2455	0.1302	0.2167	0.2256	0.2118	0.2424
*K* _3_	0.0917	0.2237	0.2341	0.1849	0.2542	0.1302	0.2372	0.2384	0.2202	0.2369
*K* _4_	0.1833	0.2329	0.2033	0.1849	0.2282	0.2604	0.2255	0.2333	0.2508	0.2507
*K* _5_	0.2750	0.2115	0.2187	0.2774	0.2484	0.2604	0.2518	0.2384	0.2536	0.2534
*K* _6_	0.2750	0.2758	0.2741	0.3698	0.2340	0.2604	0.2401	0.2384	0.2480	0.2479
*K* _7_	0.1833	0.2544	0.2495	0.0925	0.2455	0.2604	0.2547	0.2409	0.2536	0.2479
*K* _8_	0.3667	0.2421	0.2495	0.1849	0.2340	0.3906	0.2079	0.2486	0.2536	0.2452
*K* _9_	0.4583	0.2513	0.2618	0.0925	0.2571	0.2604	0.2606	0.2333	0.2648	0.2424
*K* _10_	0.1833	0.2360	0.2680	0.1849	0.2109	0.1302	0.2460	0.2384	0.2425	0.2259
*K* _11_	0.4583	0.2636	0.2187	0.5547	0.2484	0.5208	0.2694	0.2512	0.2258	0.2286
*K* _12_	0	0.1839	0.1910	0.1849	0.2282	0.2604	0.2401	0.2358	0.2285	0.2396
*K* _13_	0.1833	0.2482	0.2403	0.1849	0.2340	0.1302	0.2372	0.2384	0.2007	0.2231
*K* _14_	0.0917	0.2298	0.2341	0.2774	0.2109	0.1302	0.2196	0.2435	0.2118	0.2259
*K* _15_	0.0917	0.2666	0.2711	0.1849	0.2051	0.1302	0.2313	0.2307	0.2174	0.2369
*K* _16_	0.2750	0.2452	0.2495	0.0925	0.2051	0.1302	0.2050	0.2281	0.2258	0.2093
*K* _17_	0.0917	0.2329	0.2433	0.1849	0.2369	0.1302	0.2167	0.2102	0.2480	0.2149
*K* _18_	0.0917	0.2452	0.2526	0.1849	0.2484	0.1302	0.2284	0.2333	0.2341	0.2204
	*C* _11_	*C* _12_	*C* _13_	*C* _14_	*C* _15_	*C* _16_	*C* _17_	*C* _18_	*C* _19_	*C* _20_
*K* _1_	0.2394	0	0.1864	0.2364	0.2534	0.1205	0.1254	0.2074	0.1859	0.1847
*K* _2_	0.2367	0.2182	0.0559	0.2117	0.2479	0.0844	0.1500	0.1037	0.0465	0.2771
*K* _3_	0.2234	0.2182	0.2610	0.2474	0.2369	0.1567	0.1570	0.3111	0.3408	0.0924
*K* _4_	0.2474	0	0	0.2364	0.2148	0.1567	0	0.2074	0.2014	0.1847
*K* _5_	0.2447	0.2182	0.0932	0.2007	0.2258	0.1326	0.0824	0.1037	0.0775	0.3694
*K* _6_	0.2527	0	0.2610	0.2529	0.2203	0.1687	0.1196	0.3111	0.3718	0.2771
*K* _7_	0.2420	0.4364	0.2610	0.2447	0.2479	0.1446	0.2014	0.2074	0.1859	0.3232
*K* _8_	0.2341	0.4364	0.2610	0.2254	0.2561	0.1205	0.2366	0.2074	0.0620	0.3694
*K* _9_	0.2553	0.2182	0.2610	0.2557	0.2616	0.1687	0.2376	0.3111	0.3563	0.2771
*K* _10_	0.2420	0.4364	0.2610	0.2364	0.2093	0.1205	0.1693	0.2074	0.1549	0.2309
*K* _11_	0.2474	0.2182	0.2610	0.1705	0.2561	0.0964	0.2226	0.1037	0	0.2309
*K* _12_	0.2367	0.2182	0.2610	0.2474	0.2396	0.8678	0.1861	0.3111	0.3408	0.1847
*K* _13_	0.2341	0	0.2610	0.2282	0.2258	0.0964	0.2686	0.2074	0.2324	0.1847
*K* _14_	0.2234	0.2182	0.2610	0.2447	0.2258	0.0723	0.3220	0.2074	0.1239	0.1847
*K* _15_	0.2154	0	0.2610	0.2502	0.2369	0.1085	0.3021	0.2074	0.3099	0.1385
*K* _16_	0.2208	0.2182	0.2610	0.2447	0.2369	0.1205	0.1254	0.2074	0.1859	0.1847
*K* _17_	0.2128	0	0.2610	0.2364	0.2121	0.0844	0.1500	0.1037	0.0465	0.2771
*K* _18_	0.2287	0.2182	0.2610	0.2557	0.2258	0.1567	0.1570	0.3111	0.3408	0.0924
	*C* _21_	*C* _22_	*C* _23_	*C* _24_	*C* _25_	*C* _26_	*C* _27_	*C* _28_	*C* _29_	*C* _30_
*K* _1_	0.3030	0.2425	0.2554	0.2405	0.2117	0.2020	0.2378	0.2352	0.2361	0.2433
*K* _2_	0.3367	0.2425	0.2554	0.2352	0.3529	0.2105	0.2326	0.2326	0.2415	0.2377
*K* _3_	0.2694	0.2425	0.2128	0.2193	0.2823	0.2077	0.2352	0.2378	0.2524	0.2263
*K* _4_	0.3030	0.2425	0.2554	0.2378	0.1059	0.2418	0.2456	0.2275	0.2442	0.2433
*K* _5_	0.3704	0.2425	0.2554	0.2484	0.0353	0.2361	0.2352	0.2326	0.2415	0.2377
*K* _6_	0.3030	0.2425	0.2554	0.2458	0.1764	0.2504	0.2430	0.2378	0.2469	0.2263
*K* _7_	0.2694	0.2425	0.1277	0.2458	0	0.2447	0.2456	0.2378	0.2334	0.2433
*K* _8_	0.2020	0.2425	0.2554	0.2326	0.1764	0.2304	0.2404	0.2429	0.2307	0.2518
*K* _9_	0.2357	0.2425	0	0.2537	0	0.2447	0.2482	0.2454	0.2415	0.2490
*K* _10_	0.2357	0.2425	0.2554	0.2220	0.1764	0.2219	0.2378	0.2429	0.2469	0.2575
*K* _11_	0.2020	0	0.2128	0.2114	0.0706	0.2333	0.2378	0.2454	0.2225	0.2603
*K* _12_	0.2020	0.2425	0.2554	0.2405	0.2117	0.2248	0.2430	0.2429	0.2334	0.2546
*K* _13_	0.1684	0.2425	0.2554	0.2352	0.2823	0.2475	0.2169	0.2326	0.2171	0.2179
*K* _14_	0.1347	0.2425	0.2554	0.2220	0.3529	0.2361	0.2064	0.2301	0.2171	0.2235
*K* _15_	0.1347	0.2425	0.2554	0.2114	0.4588	0.2447	0.2195	0.2352	0.2334	0.2009
*K* _16_	0.1010	0.2425	0.2128	0.2484	0.1059	0.2504	0.2456	0.2275	0.2388	0.2179
*K* _17_	0.1010	0.2425	0.2554	0.2352	0.2117	0.2560	0.2352	0.2250	0.2198	0.2263
*K* _18_	0.1010	0.2425	0.2554	0.2511	0.3529	0.2504	0.2326	0.2301	0.2415	0.2150

**Table 7 ijerph-19-10607-t007:** The weights.

Weights	Tertiary Indicators									
	*C* _1_	*C* _2_	*C* _3_	*C* _4_	*C* _5_	*C* _6_	*C* _7_	*C* _8_	*C* _9_	*C* _10_
subjective weights	0.0358	0.0356	0.0360	0.0327	0.0349	0.0348	0.0347	0.0358	0.0339	0.0343
objective weights	0.0774	0.0149	0.0163	0.0609	0.0087	0.0621	0.0090	0.0045	0.0091	0.0065
comprehensive weight	0.0566	0.0253	0.0262	0.0468	0.0218	0.0485	0.0218	0.0202	0.0215	0.0204
	*C* _11_	*C* _12_	*C* _13_	*C* _14_	*C* _15_	*C* _16_	*C* _17_	*C* _18_	*C* _19_	*C* _20_
subjective weights	0.0336	0.0333	0.0341	0.0365	0.0360	0.0318	0.0320	0.0318	0.0322	0.0332
objective weights	0.0062	0.0999	0.0439	0.0109	0.0081	0.1351	0.0560	0.0384	0.0666	0.0412
comprehensive weight	0.0199	0.0666	0.0390	0.0237	0.0221	0.0566	0.0253	0.0262	0.0468	0.0218
	*C* _21_	*C* _22_	*C* _23_	*C* _24_	*C* _25_	*C* _26_	*C* _27_	*C* _28_	*C* _29_	*C* _30_
subjective weights	0.0305	0.0313	0.0299	0.0293	0.0356	0.0338	0.0287	0.0338	0.0325	0.0315
objective weights	0.0454	0.0294	0.0339	0.0066	0.0783	0.0080	0.0055	0.0032	0.0053	0.0084
comprehensive weight	0.0485	0.0218	0.0202	0.0215	0.0204	0.0199	0.0666	0.0390	0.0237	0.0221

**Table 8 ijerph-19-10607-t008:** The matrix after weighted re-standardization and the optimal solution distance of 18 communities.

Communities	Weighted Standardization	The Optimal Solution Distance
*K* _1_	0.1839	0.0054
*K* _2_	0.1937	0.0218
*K* _3_	0.2149	0.0574
*K* _4_	0.1807	0
*K* _5_	0.2059	0.0422
*K* _6_	0.2346	0.0904
*K* _7_	0.2204	0.0666
*K* _8_	0.2458	0.1091
*K* _9_	0.2320	0.0861
*K* _10_	0.2197	0.0655
*K* _11_	0.2300	0.0826
*K* _12_	0.2740	0.1563
*K* _13_	0.1964	0.0264
*K* _14_	0.2063	0.0430
*K* _15_	0.2056	0.0417
*K* _16_	0.2046	0.0401
*K* _17_	0.1823	0.0027
*K* _18_	0.2181	0.0627

## Data Availability

The data are available from the corresponding author upon reason able request.

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
