# Peer review of "Construction and Evaluation of a Safe Community Evaluation Index System—A Study of Urban China"

_ijerph, 2022, doi:10.3390/ijerph191710607_

Round 1

Reviewer 1 Report

This paper constructs a framework to evaluate a safe community and applied the framework to 18 communities in Xi’an, China. I can easily find that the paper was written on authors’ great efforts for research, and I appreciate its potential novelty to academic and practical fields.

However, the paper needs more logical explanation to connect concepts and sophistication in academic paper writing styles. Please read my comments and questions below and revise, or give counter-comments if needed.

Whole paper:

->Please define “safe community.” Is it a general term or academic terminology? And what is its relations with community resilience? World Health Organization (WHO) also uses the term “Safe Community” as a different concept from authors’, so the authors need to make the meaning of “safe community” clear.

This study is based on the previous study conducted by Gao Feng and Zhu Yuguo which discussed the sense of community security. On page 4, the authors stated “the concept of community security is defined as the collective consciousness of community residents, ... but also the cognition of the impact of social policies at the macro and micro levels on community residents, .... The three levels of consciousness, technology, and policy are the starting point for constructing the theoretical model of community resilience from the perspective of security.” As far as I understand, the sense of community security is subjective issues while the tertiary indicators include objective variable on table 1. Relations among safe community, sense of security and the tertiary indicators are not clear. The authors need to explain how they are connected and how the study is related to the previous studies especially the one of Gao Feng and Zhu Yuguo. And why the sense of community security is important to the safe community and community disaster resilience also should be discussed. The present paper lacks its conceptual picture and readers might get lost for what the study intends to contribute to.

p.1, l.22: Abstract

->Sentences misguides readers. The first result is not by the analysis but based on the literature review and the initial evaluation index sifting. Please indicate by which methodologies what findings were found. Otherwise, all of the results from (1) to (4) seem to get from the evaluation results.

p.1, l.45: “Compared with the traditional concept of public security defence, a safe community can better adapt to changes in the external environment, which are characterized by the high uncertainty, low predictability, and high destructiveness of emergencies, to realize the safe and sustainable development of the community.”

-> Please explain why a safe community can better adapt than the traditional concept with references. 

p.3, l.129: “most community resilience index systems constructed by current research have certain defects in the empirical validity and reliability of the method.”

-> The authors need to explain what the defects are and how this can be overcome by the present study.

p.3, l.147: “According to previous research on X and Y communities”

-> Please indicate which previous study the authors mean.

p.4, l.165, 174, 179:

-> Authors explain definitions of consciousness, technology, and policy. But there is no reference. If they are not your opinions or findings, please make sure to refer in a proper way.

p.4, l.182: “previous research”

-> Please indicate which previous study the authors mean.

p.4, 2.2 Initial evaluation index sifting:

-> As this section is a part of the present study, authors need to write details of questionnaire survey and interviews with relevant experts in academia and industry (5W1H: why, what, where, when, to whom, and how).

p.8, Figure 1:

-> I could not get how the authors connect 5 primary indicators with 4 common factors. The authors found four common factors but how they are reflected to the Figure 1? What are purposes of the explanatory factor analysis?

p.8-, 3. Construction of a safe community assessment model based on the TOPSIS method

-> Some tertiary indicators can be understood while I could not grasp others. For example, C25 is a converted indicators but the accumulation of funds should be converted? And C13 (refuge), C22 (building density), C23 (the greening rate) are an interval indicator but I do not get why and/or how. The same thing is applicable to other indicators (such that what C5 [community post-disaster summary] means; how C17 [population density] positively affect the safe community, etc.). Please explain how and why each tertiary indicator should be treated for analysis.

p.11, Figure 2:

-> Is it correct that “Calculate subjective weights” comes after “Calculate subjective weights” and “Calculate objective weights”?

p.11, l.342: “Six districts in Xi'an are selected”

-> Please indicate why Xi’an and six districts were selected according to the present study’s purpose. 

p.11, l.344: “The selection principles of representative communities are as follows: First, select a large community with a permanent population of more than 30000; Secondly, select an old community built in the 1980s; Finally, select a new community built after 2015.”

-> Please indicate why the authors chose these three types of communities for the present study’s purpose.

p.11, l.356: “For example, indicators ...”

-> I could not understand the relations of this sentence and the previous sentence. They seem to indicate different things.

p.14, l.388: “results recognized in the industry”

-> I could not get why the authors chose the industry to confirm the effectiveness and reliability. Please clearly indicate who they are and how they are appropriate to the confirmation.

Author Response

  • Comment 1: This paper constructs a framework to evaluate a safe community and applied the framework to 18 communities in Xi’an, China. I can easily find that the paper was written on authors’ great efforts for research, and I appreciate its potential novelty to academic and practical fields. However, the paper needs more logical explanation to connect concepts and sophistication in academic paper writing styles. Please read my comments and questions below and revise, or give counter-comments if needed.

Authors’ Response: We thank the referee for the positive evaluation of our work. We have carefully read all the comments and questions by the review experts, and we revised the manuscript in detail according to the suggestions of experts. The manuscript has been greatly improved, making the connection between theory and practice in academic paper writing styles. Thank you again for your review.

  • Comment 2: Please define “safe community.” Is it a general term or academic terminology? And what is its relations with community resilience? World Health Organization (WHO) also uses the term “Safe Community” as a different concept from authors’, so the authors need to make the meaning of “safe community” clear.

Authors’ Response: Thank you for your suggestion. Your suggestion make our manuscript more complete and detailed. Considering the structure of the manuscript, we added the concept analysis of “safe community” and “resilient community” in Section 2.1. See Page 5(Lines 209-224). The major revision are as follows:

“Resilience community” is the latest urban governance concept. Resilience is not a single concept but rather a generalization of a system framework. The purpose of community construction under the framework of this system is to improve the community’s ability to respond positively to the crisis, adaptability and sustainable development. Through on-the-spot investigation and resident interviews within the X and Y communities in Xi’an, Shaanxi Province. Our community residents are not only concerned about the infrastructure construction of disaster prevention and mitigation, but also macro and microlevel policies can attract more attention from community residents than community security issues.

Accordingly, we define the concept of community security as the collective consciousness of community residents, including not only the security and feelings of community residents themselves but also the cognition of the impact of social policies at the macro and micro levels on community residents, their families and even the whole community. According to this, we build the theoretical model of safe community from the three aspects of “consciousness, technology and policy”. The specific conceptual relationship is shown in Figure 1.

  • Comment 3: This study is based on the previous study conducted by Gao Feng and Zhu Yuguo which discussed the sense of community security. On page 4, the authors stated “the concept of community security is defined as the collective consciousness of community residents, ... but also the cognition of the impact of social policies at the macro and micro levels on community residents, .... The three levels of consciousness, technology, and policy are the starting point for constructing the theoretical model of community resilience from the perspective of security.” As far as I understand, the sense of community security is subjective issues while the tertiary indicators include objective variable on table 1. Relations among safe community, sense of security and the tertiary indicators are not clear. The authors need to explain how they are connected and how the study is related to the previous studies especially the one of Gao Feng and Zhu Yuguo. And why the sense of community security is important to the safe community and community disaster resilience also should be discussed. The present paper lacks its conceptual picture and readers might get lost for what the study intends to contribute to.

Authors’ Response: Thank you for your suggestion. Your suggestion make our manuscript more complete and detailed. The major revision are as follows:

(1) We added the concept analysis of “safe community” and “resilient community” in Section 2.1. See Page 5(Lines 209-224).

(2) We added the conceptual picture in Page 5(Figure 1).

(3) This study is based on the previous study conducted by Gao Feng and Zhu Yuguo which discussed the sense of community security. In order to distinguish which tertiary indicators are derived from the existing literature summary and which are derived from our research results. In Table 1, we added the references of relevant literatures. What we did not note was the tertiary indicators that we discussed and researched results. See Page 6-7.

  • Comment 4: 1, l.22: Abstract

Sentences misguides readers. The first result is not by the analysis but based on the literature review and the initial evaluation index sifting. Please indicate by which methodologies what findings were found. Otherwise, all of the results from (1) to (4) seem to get from the evaluation results.

Authors’ Response: Thank you for your suggestion. This is our negligence in expression. The manuscript is checked again improve its expression and grammar by American Journal Experts (No. CV1C127X). The major revision are as follows:

Abstract: A community is the basic unit of a city. The scientific and effective evaluation of the construction effect of safe communities can improve the construction capacity of community disaster prevention and mitigation; it is also the basis for improving urban public safety and realizing stable and sustainable urban operation. First, following the development framework of a safe community and taking two typical communities in Xi’an, China, as examples, based on the literature and expert opinions, the initial indicators of a safe community are determined. Second, based on existing data, the literature and expert opinions, a questionnaire is designed, and the reliability and validity of the questionnaire are tested by exploratory factor analysis. Third, the indicators for evaluating the construction ability of a safe community are selected. Finally, the evaluation model of the construction ability of safe communities is constructed by using the comprehensive weighting technique for order of preference by similarity to the ideal solution (TOPSIS), which is applied to the actual evaluation of eighteen representative communities in Xi’an. The main findings are as follows. (1) The sense of community security is the collective consciousness of community residents. It includes not only the security and feelings of community residents themselves but also the cognition of the impact of social policies at the macro and micro levels on community residents, their families and even the whole community. (2) From the three levels of consciousness, technology, and policy as the starting points for the construction of the theoretical model of a safe community, organizational resilience, accessibility resilience, social environmental resilience and capital resilience are found to be the main influencing factors in the construction of a safe com-munity. (3) Using questionnaires and expert interviews to preliminarily screen evaluation indicators and using the comprehensive weighting TOPSIS method to build an evaluation model can effectively avoid the defects of traditional empirical research on the validity and reliability of methods. (4) The ranking of the eighteen representative communities in the empirical analysis is basically consistent with the selection results of the national comprehensive disaster reduction demonstration community, which indicates the effectiveness and accuracy of the indicators and algorithms.

  • Comment 5:1, l.45

“Compared with the traditional concept of public security defence, a safe community can better adapt to changes in the external environment, which are characterized by the high uncertainty, low predictability, and high destructiveness of emergencies, to realize the safe and sustainable development of the community.” Please explain why a safe community can better adapt than the traditional concept with references.

Authors’ Response: Thank you for your comment. This is our negligence in expression. In order to avoid ambiguity, we deleted this sentence and added the concept and characteristics of “safe community” to the manuscript in Section 2.1. See Page 5(Lines 209-224).

  • Comment 6:3, l.129

“most community resilience index systems constructed by current research have certain defects in the empirical validity and reliability of the method.” The authors need to explain what the defects are and how this can be overcome by the present study.

Authors’ Response: Thank you for your suggestion. Your suggestions make our manuscript more complete and detailed, and the combination of theory and practice more perfect. We added the explain what the defects are and how this can be overcome by the present study in Page 3-4 (Lines 150 to 189). The major revision are as follows:

There are mainly three aspects.

First, most of the safe community indicator systems are constructed by means of literature, expert interviews and questionnaires, without examining the corrected item total correlation (CITC) of each indicator, especially verifying that the indicator is deleted. If the indicator is deleted, α coefficient is significantly increased, this indicates that the indicator is not very representative of the construction of safe communities and should thus be deleted. This study verifies the CITC of 35 initial tertiary indicators screened through the literature and expert interviews, as well as the 35 α coefficients after deleting the 35 initial tertiary indicators. If the indicator with a CITC less than 0.5 is deleted, the α coefficient will be greatly improved. This indicates that the indicator with a CITC of less than 0.5 cannot represent the safe community well and should thus be deleted, thereby enhancing the ac-curacy of the index construction.

Second, in terms of weight setting, the analytic hierarchy process (AHP) is mostly used in most safety community studies. Although this method is simple and practical, it relies heavily on expert scoring and is highly subjective. Especially in situations consisting of too many indicators, large data statistics and uncertain weights, the analysis matrix is too large to be solved. Through the comprehensive weighting TOPSIS method, the subjective and objective weights are comprehensively weighted to weaken the influence of subjective factors on the results to enhance the reliability and accuracy of the study.

Finally, research on safe community indicators in China has just started. The only research focuses on the perspective of management, thus, focus on related issues, such as how to build the measurement and evaluation system of community resilience, is rare. In view of the fact that the current index system for evaluating safe communities in China mostly considers the hardware infrastructure for disaster prevention and mitigation, and little or no consideration is given to the residents’ safety feelings and the impact of relevant policies such as employment, medical care, further education and old-age care on the residents, the analysis is not systematic enough, and the impact of different factors on safety community construction is not given much attention. Therefore, the index system constructed by the questionnaire, field investigation and interview methods considers the relevant safety feelings of residents, thus enhancing the practicability of the safety com-munity evaluation index system and providing a scientific basis for decision-making for managers and enriching management countermeasures.

Accordingly, this research follows the framework of the resilient development of safe communities, takes typical communities in Xi’an, China, as actual cases, uses questionnaires, expert interviews, factor analysis and the comprehensive weighting TOPSIS method to screen evaluation indicators and build evaluation models to find the main influencing factors in building a safe community; avoids the defects in the validity and reliability of traditional empirical research, and improves the ability of cities to respond to, deal with and recover from emergencies.

  • Comment 7: 3, l.147:

“According to previous research on X and Y communities” Please indicate which previous study the authors mean.

Authors’ Response: Thank you for your comment. It is indeed a question of our presentation. We added the expression in Page 4 (Lines 197 to 201). The major revision are as follows:

Through on-the-spot investigation and resident interviews within the X and Y com-munities in Xi’an, Shaanxi Province. This research of the X and Y communities was funded by the China Postdoctoral Science Foundation (grant number: 2020M673462). We find that the sense of community security can have a broader meaning; i.e., community residents pay attention not only to community security but also to the content of social policies, such as the education and employment of children around the community, the charging of medical thresholds near the community, community building facilities, hidden dangers and environmental safety, community pollution issues and other policies closely related to the production and life of community residents, the city’s medical insurance and pension policies, the city’s unemployment and employment policies, the city’s environmental pollution and other livelihood engineering issues. Macro and microlevel policies can attract more attention from community residents than community security issues.

  • Comment 8: 165, 174, 179

 Authors explain definitions of consciousness, technology, and policy. But there is no reference. If they are not your opinions or findings, please make sure to refer in a proper way.

Authors’ Response: Thank you for your suggestion. Your suggestion makes my manuscript more complete and free of intellectual property disputes. The definitions of consciousness, technology and policy are our opinions and findings. We added the expression in Page 5 (Lines 209 to 224). The major revision are as follows:

“Resilience community” is the latest urban governance concept. Resilience is not a single concept but rather a generalization of a system framework. The purpose of community construction under the framework of this system is to improve the community’s ability to respond positively to the crisis, adaptability and sustainable development. Through on-the-spot investigation and resident interviews within the X and Y communities in Xi’an, Shaanxi Province. Our community residents are not only concerned about the infrastructure construction of disaster prevention and mitigation, but also macro and microlevel policies can attract more attention from community residents than community security issues.

Accordingly, we define the concept of community security as the collective consciousness of community residents, including not only the security and feelings of community residents themselves but also the cognition of the impact of social policies at the macro and micro levels on community residents, their families and even the whole community. According to this, we build the theoretical model of safe community from the three aspects of “consciousness, technology and policy”. The specific conceptual relationship is shown in Figure 1.

  • Comment 9: 4, l.182

 “previous research” Please indicate which previous study the authors mean.

Authors’ Response: Thank you for your comment. This is our negligence. We added the expression in Page 6 (Lines 245 to 248). The major revision are as follows:

For example, through on-the-spot investigation and resident interviews within the X and Y communities in Xi’an, Shaanxi Province, we found that community residents are concerned about policies related to children’s education and employment, the city’s medical insurance and pension policies, and the city’s unemployment and employment policies.

  • Comment 10: 4, 2.2 Initial evaluation index sifting

 As this section is a part of the present study, authors need to write details of questionnaire survey and interviews with relevant experts in academia and industry (5W1H: why, what, where, when, to whom, and how).

Authors’ Response: Thank you for your suggestion. Your suggestion makes my manuscript more complete and free of intellectual property disputes. Part of the screening of the initial indicators in Table 1 is from the existing literatures and expert interviews, and the other part is from our team’s discussions. We added the details of questionnaire survey and interviews with relevant experts in academia and industry in Page7-9 (Lines 266 to 335). The major revision are as follows:

2.3.1. Preparation and distribution of questionnaires

To verify the reliability and accuracy of the 5 primary indicators, 13 secondary indicators and 35 tertiary indicators that were screened out, we designed a questionnaire that contained the initial 35 indicators, and the questionnaire was divided into two parts. The first part of the survey consisted of questions that focus on the basic personal information of community residents and community managers, including their gender, age, educational level, type of housing, building form of housing in the community, building age, and time living in the community. The second part uses a Likert scale to measure satisfaction, with answers ranging from very satisfied (5 points) to satisfied (4 points), indifferent (3 points), dissatisfied (2 points), and very dissatisfied (1 point).

The questionnaire was distributed to the community residents and community managers located in communities X and Y in Xi’an, the community staff members in charge of the emergency departments, and the professional scholars working in this field at universities located in Xi’an. For community residents and community managers, 340 questionnaires were distributed within a week by using the Questionnaire Star and WeChat apps. For the community staff in charge of the emergency departments and the experts and scholars working in this field at universities located in Xi’an, a total of 20 questionnaires were distributed within a week by using the Questionnaire Star and WeChat apps. For elderly individuals over 60 years old in the community, a paper version of the questionnaire was used,40 questionnaires were distributed within a week, and random sampling was used to find participants in the community.

The X community was established in 2000.The total area of the jurisdiction is 0.6 square kilometres, with 4416 permanent residents and a population of 11700. There are 21 courtyards and 5 units. There are 15 community work service personnel, 8 public welfare posts, 1 Party branch, 3 Party branch members, 169 Party members, and 7 on-the-job Party members; there are 8 members of the neighbourhood committee and 3 full-time community members (cross employment). The office area of the community is 336 square metres. There is a one-stop service hall, a Party building activity room, a reading room and an activity room for middle-aged and elderly individuals in the community, which can provide various convenience services for community residents, such as Party member services, labour security, and floating population management. The X community adheres to the principle of harmonious community construction, takes serving community residents as the core, and takes sustainable development of residents’ autonomy and strengthening neighbourhood harmony as the goal. It integrates resources and establishes distance education broadcasting points, popular science universities, population-based schools, staff bookstores and other characteristic lectures to disseminate knowledge and improve the quality of the masses. The community hosts parties on cool evenings, community sports games, Spring Festival couplets, Chongyang saozi noodles events, the Winter Solstice Dumpling Banquet and other activities to tighten the relationship between the Party and the masses and that between the cadres and the masses. To improve residents’ level of happiness, grid management and the dean system are implemented, and work efficiency is advocated. This approach presents a new outlook of “residents’ autonomy, orderly management, perfect service, good public order, beautiful environment, civilization and harmony”. The X community has won the honorary titles of a national disabled community rehabilitation demonstration site, a Shaanxi four-star community party organization, a Shaanxi safe family demonstration community and so on.

The Y community was rated as an affordable housing project by the Xi’an municipal government in 2004. It covers an area of 0.11 square kilometres. The total construction area of the community is more than 200000 square metres. There are 2719 permanent residents and population of 9327 people. There are kindergartens, open clubs, supermarkets, clinics and gyms in the community, of which the kindergartens cover an area of 1500 square metres and the open clubs cover an area of 1900 square metres. There are seven educational institutions positioned around the community. The water supply and power supply are under the centralized management of the municipal government, and all residential heating pipes are designed and installed in a household-by-household manner. This is not only conducive to household measurement after the public network transformation but can also regulate the indoor heating temperature. There are 10 community work and service personnel in Y community, and a Party branch organization has been established. The community has an office area of 253 square metres. The office area has a variety of functions, such as a one-stop service hall, a Party building room, and an activity room for elderly individuals. This area helps not only to serve community residents but also to maintain the community environment; it also provides more labour security, civil affairs assistance, family planning and other convenient services for community residents. As the community is located close to Cultural Park, it has more coverage of green plants, which makes the community appear more humanized.

A total of 400 questionnaires were distributed. After eliminating the invalid questionnaires, 374 valid questionnaires were finally obtained, with an effective recovery rate of 93.5%. Descriptive statistical analysis of the samples is shown in Table 2.

Table 2 Descriptive statistical analysis of samples.

Basic information

Mean

Standard

Deviation

Frequency

Gender

male

1.59

0.492

152

40.6%

female

222

59.4%

Age

Under 18 years old

4.03

0.930

2

0.5%

19-29 years old

95

25.4%

30-39 years old

206

55.1%

40-49 years old

45

12%

50-59 years old

16

4.3%

60-69 years old

7

1.9%

70 years old and above

3

0.8%

Education level

High school and below

2.13

0.581

42

11.2%

College/undergraduate

242

64.7%

Postgraduate and above

90

24.1%

House type

Commercial housing

1.56

1.952

310

82.9%

Housing reform

16

4.3%

Stock house

5

1.3%

Fund raising house

11

2.9%

Housing project

10

2.7%

Affordable housing

22

5.9%

Building form of community house

Low-rise residence (building height less than 3 floors)

3.53

0.905

62

16.6%

Multi-storey residence (3-6 floors high)

24

6.4%

Medium and high-rise residential buildings (7-9 floors high)

268

71.7%

High-rise residence (the building height is more than 10 floors)

8

2.1%

other

12

3.2%

Community building year

2000 and before

2.35

0.705

41

11%

2001-2010

169

45.2%

2011-2020

155

41.4%

2021 present

9

2.4%

Residence time

More than 10 years

2.49

0.968

49

13.1%

5-9 years

170

45.5%

1-4 years

76

20.3%

Less than 1 year

79

21.1%

  • Comment 11: 8, Figure 1

 I could not get how the authors connect 5 primary indicators with 4 common factors. The authors found four common factors but how they are reflected to the Figure 1? What are purposes of the explanatory factor analysis?

Authors’ Response: Thank you for your comment. This is our negligence in expression. We do factor analysis for purposes, and the relationship between 5 primary indicators, four factors and Figure 2 (Figure 1 has been modified to Figure 2.) is described as follows:

(1) In order to verify the CITC of the initial 35 tertiary indicators, select the tertiary indicators with CITIC less than 0.5, and check the growth of α coefficient after deleting this indicator. If these items are deleted, then the reliability coefficient will be greatly improved. It shows that the tertiary indicators can not well represent the relevant influencing factors of safe communities. See Page 9 (Lines 349-350).

(2) Our team is writing another article on factor analysis, which will be an extension and in-depth study of this manuscript. If this manuscript is accepted and published, we will cite the data and conclusions here for in-depth analysis in the paper being written.

(3) We initially screened 5 primary indicators (A1, A2, A3, A4, A5), 13 secondary indicators and 35 tertiary indicators. After the verification of a series of methods such as questionnaires, 30 tertiary indicators were selected. After factor analysis, only 4 factors (F1, F2, F3, F4) were basically in line with the expectation. The factors (F1, F2, F3, F4) here are the same as the primary indicators (A1, A2, A3, A4). In order to keep consistent with the article being written (2), we use factors (F1, F2, F3, F4) to represent the common factors.

(4) Because C28, C29, C30 tertiary indicators are important for research, especially for elderly care and disaster prevention and mitigation. C28, C29, C30 tertiary indicators cannot be classified into other primary indicators. The primary indicator A5 was retained by us in Figure 2(Figure 1 has been modified to Figure 2.). See Page 11 (Lines 374-388).

  • Comment 12:8, 3. Construction of a safe community assessment model based on the TOPSIS method

 Some tertiary indicators can be understood while I could not grasp others. For example, C25 is a converted indicators but the accumulation of funds should be converted? And C13 (refuge), C22 (building density), C23 (the greening rate) are an interval indicator but I do not get why and/or how. The same thing is applicable to other indicators (such that what C5 [community post-disaster summary] means; how C17 [population density] positively affect the safe community, etc.). Please explain how and why each tertiary indicator should be treated for analysis.

Authors’ Response: Thank you for your comment. This is our negligence. Your proposal makes my manuscript more complete and free of intellectual property disputes. We added the expression in Page 13-14 (Lines 406 to 452). The major revision are as follows:

The following describes the forward processing of different types of indicators.

(1) For maximum tertiary indicators, we do not need to perform any forward processing.

(2) For minimum tertiary indicators (the smaller the indicator value is, the better), among the 30 tertiary indicators screened in this study, C17 (population density) and C19 (the proportion of special populations) are minimum tertiary indicators. The tertiary indicator C17 (population density) is obtained from the statistical yearbook of each district. We believe that the smaller the population density is, the richer the per capita public resources are, which is relatively good. Of course, the population density cannot be infinitely small. The tertiary indicator C19 (the proportion of special populations) are obtained from the statistical yearbooks of various districts. The special population refers to disabled individuals, elderly individuals and other related groups. We believe that the smaller the proportion of the special population is, the stronger the community’s ability to resist disasters is. We convert the minimum indicators to the maximum indicators:

                                                (2)

{xi} is a group of minimum index series,  is the converted indicator.

(3) For intermediate tertiary indicators (intermediate tertiary indicators are indicators whose values should not be too large or too small, and the closer to a certain value, the better), among the 30 tertiary indicators screened in this study, there are no intermediate indicators. However, for the sake of the integrity of the study, if there are intermediate indicators, the intermediate indicators are converted into maximum indicators:

                        (3)

{xi} is a set of intermediate index series, and the best value is xbest,  is the converted indicator.

(4) For interval tertiary indicators (interval tertiary indicators are the best indicators that their values fall within a certain interval), among the 30 tertiary indicators screened in this study, C13 (refuge), C22 (building density), and C23 (the greening rate) are interval tertiary indicators. C13 (refuge) is obtained from the statistical yearbook of each district. Refuge refers to the effective per capita refuge area. According to the relevant regulations of Xi’an, the effective per capita refuge area in Xi’an is 1.5-3 m2/person, which is an interval type three-level indicator. C22 (building density) is obtained from the statistical yearbook of each district. Building density refers to the per capita effective refuge area. According to the relevant regulations of Xi’an, the building density of the Xi’an community is 20% - 39%, which means that it can maximize the use of space and public resources. C23 (the greening rate) is obtained from the statistical yearbook of each district. The greening rate is an indicator used to measure the greening degree of community roads. According to the relevant regulations of Xi’an city construction, the greening rate of the community is 20% - 30%, which makes the residents feel better. Interval tertiary indicators need to be positively processed; that is, interval tertiary indicators are converted into maximum tertiary indicators:

    (4)

{xi} is a set of intermediate index series, and the best value is [a, b].

In addition to the abovementioned tertiary indicators, i.e., C13, C17, C19, C22 and C23, the other 25 tertiary indicators are maximum tertiary indicators and do not need any forward processing. To eliminate the influence of different dimensions, the normalized index matrix is standardized, and the normalized evaluation matrix is recorded as Z.

  • Comment 13: 11, Figure 2

 Is it correct that “Calculate subjective weights” comes after “Calculate subjective weights” and “Calculate objective weights”?

Authors’ Response: Thank you for your comment. This is our mistake. I’m very sorry. Because there is one figure have been added, Figure 2 has been modified to Figure 3. See the Page 15(Lines 492-494). The major revision are as follows:

Figure 3 The algorithm flow chart.

  • Comment 14: 11, l.342

 “Six districts in Xi’an are selected” Please indicate why Xi’an and six districts were selected according to the present study’s purpose.

Authors’ Response: Thank you for your suggestion. It is indeed a question of our presentation. Your proposal makes my manuscript more complete and free of intellectual property disputes. We added the expression in Page 15-17 (Lines 495 to 552). The major revision are as follows:

This study chooses Xi’an city as the research object.

First, this research was funded by the China Postdoctoral Science Foundation (grant number: 2020M673462) and the Social Science Foundation of Shaanxi Province (grant number: 2021R028). The research objects of these two funds are communities in Xi’an.

Two, we participated in the whole process of the “Shaanxi 2021 comprehensive dis-aster reduction demonstration community evaluation”. During the evaluation, we found that most communities in Xi’an have built emergency shelters, organized emergency drills, investigated potential disaster risks, and built micro fire stations. Through the establishment of comprehensive disaster reduction demonstration communities, the community’s emergency response ability to disasters has been further improved, and the residents’ awareness of disaster prevention and reduction has also been strengthened. However, the existing indicator system of comprehensive disaster reduction demonstration community construction can no longer meet the current needs, it is urgent to increase the indicators of relevant policies that consider the residents’ safety feelings and guarantee their safety feelings. In consideration of the practical process of community disaster prevention and mitigation assessment we participated in and the problems faced by Xi’an in constructing a safe community. Therefore, Xi’an city is selected as the research object.

Selection principles and steps of representative communities in Xi’an:

(1) According to the “list of comprehensive disaster reduction demonstration com-munities in Shaanxi Province in 2021” announced by Shaanxi Provincial Emergency Management Department on December 3, 2021, there are 24 communities in Xi’an.

(2) According to the recommendation of Shaanxi Province in 2017, “42 communities in Shaanxi Province were selected as national comprehensive disaster reduction demonstration communities”; among them, 6 communities from Xi’an were selected. Since the 24 communities in (1) and the 6 communities in (2) do not all represent the same community, a total of 30 communities were screened.

(3) At present, Xi’an is divided into 11 districts and 2 counties. Since the index system of this study involves many factors such as employment, medical care, further education and old-age care on the residents, according to the distribution of resources such as further education, elderly care and employment in Xi’an, only the six districts of Xi’an where these resources are concentrated are selected as the community research objects (these six districts are Lianhu District, Beilin District, Yanta District, Xincheng District, Weiyang District and Chang’an District). To ensure the interpretability, fairness and consistency of the research results, three representative communities were selected for each district. Se-lection principles see (4)-(5).

(4) As most of the existing large-scale communities in Xi’an are “a city within a city”, they face many community safety management problems and are very valuable research objects of safe communities. The population of each district and county is published on the Xi’an Municipal People’s government website. According to the regulations on the size of the community population in China, each district selected a large community with a resident population of more than 30000.

(5) According to the statistical yearbook of Xi’an, there approximately 30% of the communities in Xi’an are old. The area of these old communities is relatively small. Block service facilities are generally shared by several communities. The construction quality is low. Most of these community residents are elderly, and their economic capacity is limited. It is a very representative community for a safe community. Therefore, each district selected an old community built prior to the 1980s.

(6) Since 2015 and especially in the past three years, to strengthen community capacity building, Xi’an started the construction of “smart communities”. These communities are not limited to “hard” infrastructure, such as living areas, traffic and the surrounding environment, but “soft” services, such as communication services, security precautions and external communication, are gradually becoming the focus of attention. It is also the development direction of the community. Therefore, each district selected a “smart community” that was built after 2015.

According to the above principles and steps, a total of 18 representative communities were selected from 30 communities, and the comprehensive weighting TOPSIS method constructed above was used to evaluate and rank the public security capacity of these 18 communities.

  • Comment 15: 11, l.344

“The selection principles of representative communities are as follows: First, select a large community with a permanent population of more than 30000; Secondly, select an old community built in the 1980s; Finally, select a new community built after 2015.” Please indicate why the authors chose these three types of communities for the present study’s purpose..

Authors’ Response: Thank you for your suggestion. This is our negligence. Your suggestions make my manuscript more complete and detailed. The answer and expression of this comment are basically consistent with Comment 14. We added the expression in Page 15-17 (Lines 495 to 552).  

  • Comment 16: 11, l.356

“For example, indicators ...” I could not understand the relations of this sentence and the previous sentence. They seem to indicate different things.

Authors’ Response: Thank you for your comment. This is our negligence. We are so sorry. We have deleted this sentence “For example, indicators ...”. See the Page 17 (Line 560).

  • Comment 17: 14, l.388

“results recognized in the industry” I could not get why the authors chose the industry to confirm the effectiveness and reliability. Please clearly indicate who they are and how they are appropriate to the confirmation.

Authors’ Response: Thank you for your suggestion. This is our negligence. Your suggestions make my manuscript more complete and detailed. We added the expression in Page 20 (Lines 587 to 602). The major revision are as follows:

We participated in the whole process of the “Shaanxi 2021 comprehensive disaster reduction demonstration community evaluation” by Shaanxi Provincial Emergency Management Department on December 3, 2021. For the 18 communities screened in this study, the ranking results of Shaanxi provincial emergency management department are as follows: K12, K8, K6, K7, K15, K9, K14, K18, K3, K10, K5, K11, K16, K13, K2, K1, K17, K4. The comparison results show that the optimal community is basically consistent with the ability results recognized in the industry.

The ranking of K9, K10, K11 and K14 communities are different. The reason is that the index system we put forward takes into account the feelings related to residents’ safety and the policies of elderly care, medical treatment, further education and employment that residents are more concerned about. We not only put forward the relevant index system but also respond with a high weight. At present, the existing safety community evaluation index system does not take these aspects into account, therefore the ranking results of these communities will be different. Security capacity building is basically consistent, which confirms the effectiveness and reliability of the selected community safety ability indicators.

Reviewer 2 Report

Congratulations on producing the work with the ideas in this paper. However, to improve this paper, we have revised several parts and are in accordance with the suggestions in the Word file attached to this review column, as follows:

1. The abstract should be a total of about 200 words maximum

2.  Please use standard words or sentences in paper script, see lines 56, 58, 82 and other words.

3.  There is still a need for studies from several previous researchers who examine relevant topics. See lines 71 and  other quotes that require encouragement from other researchers .

4.  Maybe you can mention the source of the data in the table or figure (page 5)!

5. Use more up-to-date literature, maybe the last 5-10 years (see literature 1)

Author Response

  • Comment 1: Congratulations on producing the work with the ideas in this paper. However, to improve this paper, we have revised several parts and are in accordance with the suggestions in the Word file attached to this review column.

Authors’ Response: We thank the referee for the positive evaluation of our work. The paper is checked again improve its expression and grammar by American Journal Experts (No. CV1C127X).

  • Comment 2: The abstract should be a total of about 200 words maximum.

Authors’ Response: Thank you for your comment. We try our best to compress the abstract and only keep the key information, but it still exceeds the standard of 200 words. Could we apply for permission to exceed 200 words? If not, we can only delete some key information. We are looking forward to hearing from you soon. The major revision are as follows:

A community is the basic unit of a city. The scientific and effective evaluation of the construction effect of safe communities can improve the construction capacity of community disaster prevention and mitigation; it is also the basis for improving urban public safety and realizing stable and sustainable urban operation. First, following the development framework of a safe community and taking two typical communities in Xi’an, China, as examples, based on the literature and expert opinions, the initial indicators of a safe community are determined. Second, based on existing data, the literature and expert opinions, a questionnaire is designed, and the reliability and validity of the questionnaire are tested by exploratory factor analysis. Third, the indicators for evaluating the construction ability of a safe community are selected. Finally, the evaluation model of the construction ability of safe communities is constructed by using the comprehensive weighting technique for order of preference by similarity to the ideal solution (TOPSIS), which is applied to the actual evaluation of eighteen representative communities in Xi’an. The main findings are as follows. (1) The sense of community security is the collective consciousness of community residents. It includes not only the security and feelings of community residents themselves but also the cognition of the impact of social policies at the macro and micro levels on community residents, their families and even the whole community. (2) From the three levels of consciousness, technology, and policy as the starting points for the construction of the theoretical model of a safe community, organizational resilience, accessibility resilience, social environmental resilience and capital resilience are found to be the main influencing factors in the construction of a safe com-munity. (3) Using questionnaires and expert interviews to preliminarily screen evaluation indicators and using the comprehensive weighting TOPSIS method to build an evaluation model can effectively avoid the defects of traditional empirical research on the validity and reliability of methods. (4) The ranking of the eighteen representative communities in the empirical analysis is basically consistent with the selection results of the national comprehensive disaster reduction demonstration community, which indicates the effectiveness and accuracy of the indicators and algorithms.

  • Comment 3: Please use standard words or sentences in paper script, see lines 56, 58, 82 and other words.

Authors’ Response: Thank you for your comment and letting us know about English language issues. Regarding the question about the spelling of program vs. programme, the British spelling of this word is “programme,” and the American English spelling is “program.” We would like our manuscript edited in British English, and we have revised the full text of the manuscript using British English.

  • Comment 4: There is still a need for studies from several previous researchers who examine relevant topics. See lines 71 and other quotes that require encouragement from other researchers.

Authors’ Response: Thank you for your comment. We re searched the literatures for nearly 5-10 years and added relevant information to the manuscript in Page 2 (Lines 60-93). The major revision are as follows:

Singh et al. [2] proposes a framework aimed at quantifying the disaster resilience of urban systems while ensuring an adequate level of sustainability, all according to a social and human-centric perspective, urban networks are modelled as hybrid social–physical networks (HSPNs) by merging both physical and social components, and engineering measures are performed on HSPNs as a measure of urban efficiency within a multiscale approach. Some scholars indicate that social indicators are identified to characterise quality of life in the aftermath of a catastrophic event [3-4]. Both efficiency and quality of life indicators are evaluated using a time–discrete approach before and after an extreme event occurs and during the recovery phase to measure inhabitant happiness and environmental sustainability [5-6]. Some scholars have also studied the resilience of communities to flood resistance [7-9], community resilience to earthquakes [10-11], community resilience to windstorms [12-13], and the methodology for evaluating community resilience [14-16].

The construction of resilient cities in China has just begun, and the construction of resilient communities has not attracted enough attention. There is room for improvement in many aspects, such as the establishment of resilient community agenda, community public spaces and community governance [18].

Huang [20], based on a case study of an old settlement of the Kucapungane (Rukai) people in Taiwan who experienced a forced relocation driven by the 2009 Typhoon Morakot, argued that heritage preservation serves as a link connecting the past and the future, through which communities have a better chance to orient themselves in navigating dis-placement and participating in postdisaster recovery.

  • Comment 5: Maybe you can mention the source of the data in the table or figure (page 5)!

Authors’ Response: Thank you for your suggestion. Your proposal makes my manuscript more complete and free of intellectual property disputes. Part of the screening of the initial indicators in Table 1 is from the existing literatures and expert interviews, and the other part is from our team’s discussions. The major revision are as follows:

(1) We have marked the cited references below Table 1. See Page 7 (Lines 263-264)

(2) Figure 2 is a diagram of the final indicators screened in Table 1 through questionnaire analysis and verification, which has been explained below Figure 2. See Page 11 (Lines 387-388).

  • Comment 6: Use more up-to-date literature, maybe the last 5-10 years (see literature 1)

Authors’ Response: Thank you for your suggestion. We re searched the literatures for nearly 5-10 years and added relevant information to the manuscript in Page 21. The major revision are as follows:

References:

[2] Singh, P.; Salmon, P.; Goode, N.; Gallina, J. Translation and evaluation of the Baseline Resilience Indicators for Communities on the Sunshine Coast, Queensland Australia. International Journal of Disaster Risk Reduction 2014, 10, 116-126.

[3] Maru, Y. T.; Stafford, S. M.; Sparrow, A.; Pinho, P. F.; Dube, O. P. A linked vulnerability and resilience framework for adaptation pathways in remote disadvantaged communities. Global Environmental Change 2014, 28, 337-350.

[4] Bozza, A.; Asprone, D.; Manfredi, G. Developing an integrated framework to quantify resilience of urban systems against disasters. Natural Hazards 2015, 78, 1729-1748.

[5] Rgodschalk, D.; Xu, C. Urban hazard mitigation: Creating resilient cities. Natural Hazards Review 2015, 4, 136-143.

[6] Alshehri, S. A.; Rezgui, Y.; Li, H. Disaster community resilience assessment method: A consensus-based Delphi and AHP approach. Natural Hazards 2015, 1, 395-416.

[7] Qasim, S.; Qasim, M.; Shrestha, R. P.; Khan, A. N.; Tun, K. Community resilience to flood hazards in Khyber Pukhthunkhwa province of Pakistan. International Journal of Disaster Risk Reduction 2016, 18, 100-106.

[8] Fulvio, T.; Francesco, R.; Fauto, M. Adapting and Reacting to Measure an Extreme Event: A Methodology to Measure Disaster Community Resilience. Energy Procedia 2016, 95, 491-498.

[9] Unnikrishnan, V. U.; Michele, B. Performance-Based Comparison of Different Storm Mitigation Techniques for Residential Buildings. Journal of Structural Engineering 2016, 142, 04016011.

[10] Sutley, J. E.; Lindt, V.D.; John, W.; Peek, L. Community-Level Framework for Seismic Resilience. I: Coupling Socioeconomic Characteristics and Engineering Building Systems. Natural Hazards Review 2017, 18, 04016014.

[11] Hasan, M. H.; Kadir, S. B. Social Assessment of Community Resilience to Earthquake in Old Dhaka. Natural Hazards Review 2020, 21, 05020004.

[12] Hassan, M.; Ameri, M. R.; Van, D. Wind Performance Enhancement Strategies for Residential Wood-Frame Buildings. Journal of Performance of Constructed Facilities 2018, 32, 04018024.

[13] Koliou, M.; Lindt, J. Development of Building Restoration Functions for Use in Community Recovery Planning to Tornadoes. Natural Hazards Review 2020, 21, 04020004.

[14] Ceskavich, R.; Sasani, M. Methodology for Evaluating Community Resilience. Natural Hazards Review 2018, 19, 04017021.  

[15] Kelman, I.; Ahmed, B. Measuring Vulnerability to Environmental Hazards: Qualitative to Quantitative. Springer Nature: Springer, Cham, Switzerland, 2020; pp. 421-452.

[16] William, H.; Zhang, W.; Ding, Z.X.; Li, X. Integrated Structural and Socioeconomic Hurricane Resilience Assessment of Residential Buildings in Coastal Communities. 2022, 23, 04022017.

[18] Liao, M. L.; Su, Y.; Li, F. F. Urban Community Construction under Framework of Resilience System. Chinese Public Administration 2018, 394, 59-64.

[20] Huang S M . Heritage and Postdisaster Recovery: Indigenous Community Resilience[J]. Natural Hazards Review, 2018, 19(4):05018008.1-05018008.12.

Round 2

Reviewer 1 Report

The paper dramatically got improved and the quality got greater with its hardworking surveys and analyses. However, there are some parts the authors need to add more explanations on which I had commented in the 1st round of the review.

To point out my comments, I refer to my previous comment numbers.

Comment 10: The authors added sufficient information on surveys for the verification of the reliability and accuracy of 5 primary indicators, 13 secondary indicators and 35 tertiary indicators. But even though it is the initial indicator identification, the authors need to add information on the second stage (cluster analysis on questionnaire survey data and network big data collected in the early stage) and the fourth stage (the opinions and suggestions of relevant experts in academician and industry), both of which are just briefly introduced on p.6. They are a part of the surveys. Or if the authors published the detail explanation in other papers or reports, etc., the authors could add a reference or information.

Author Response

Reviewer 1:

  • Comment 1: The paper dramatically got improved and the quality got greater with its hardworking surveys and analyses. However, there are some parts the authors need to add more explanations on which I had commented in the 1st round of the review.

Authors’ Response: We thank the referee for the positive evaluation of our work. We have carefully read all the comments and questions by the review expert. I'm very sorry that the revision of the Comment 10 in the first round of modification did not satisfy you. We revised the manuscript in detail according to the suggestion again. The manuscript has been greatly improved in academic paper writing styles. Thank you again for your review.

  • Comment 2: To point out my comments, I refer to my previous comment numbers. Comment 10: The authors added sufficient information on surveys for the verification of the reliability and accuracy of 5 primary indicators, 13 secondary indicators and 35 tertiary indicators. But even though it is the initial indicator identification, the authors need to add information on the second stage (cluster analysis on questionnaire survey data and network big data collected in the early stage) and the fourth stage (the opinions and suggestions of relevant experts in academician and industry), both of which are just briefly introduced on p.6. They are a part of the surveys. Or if the authors published the detail explanation in other papers or reports, etc., the authors could add a reference or information.

Authors’ Response: Thank you for your suggestion. I'm very sorry that the revision of the Comment 10 in the first round of modification did not satisfy you. Your suggestion make our manuscript more complete and detailed. In the first round of modification, the major revision are as follows:

  1. Part of the screening of the initial indicators in Table 2 is from the existing literatures and expert interviews, and the other part is from our team’s discussions. We added the details of questionnaire survey and interviews with relevant experts in academia and industry in Page7-9 (Lines 292 to 365).
  2. We added the details of descriptive statistical analysis of samples in Page 8-9. See Table 2 (Lines 364 to 365).

In the second round of modification, we added the details of the process of selecting the initial indicators of safe communities in Page 6-7 (Lines 254-291). The major revision are as follows:

1.  2.2 Initial evaluation index sifting

Based on the analysis of the previous literature, the selection of initial indicators of safe communities follows the five principles of scientificity, representativeness, operability, applicability and comprehensiveness. The process of selecting the initial indicators of safe communities was divided into the following four steps.

First, selection was based on the induction and analysis of the relevant literature and practical cases in China and other countries. Marked tertiary indicators in Table 2 are from the existing literatures and expert interviews, and the other part is the discussion and research of our team.

Second, cluster analysis on questionnaire survey data and network big data collected in the early stage was carried out to screen the influencing factors that affect the safety feelings of community residents. The data source includes two parts, one part is from the survey data: including from the questionnaire, expert interviews and historical data collation; The other part comes from big data on the Internet: including "local treasure" in Xi'an city and "community owners forum" and " WeChat apps group" in X community and Y community, as well as other relevant web pages, forums and WeChat apps. It mainly completes the following three aspects of work. (1) Supplement the lack of questionnaire and expert interview data, so as to obtain accurate and real-time data and expand the data set; (2) Discover the change and evolution of the residents' mentality in the face of specific disasters, and provide scientific data support for the final policy recommendations; (3) Provide help for building community resilience indicators. Obtain relevant data information such as user publishing time, emergency keywords, emotion words and place words from the network. The key data summary obtained from these relevant web pages and forums is shown in Table 1.

Table 1 Key data of relevant web pages and forums.

Keywords

Time

Place

Emotion words

Selection reason

determine the keywords of the emergency

determine the time node of data crawling

determine the data acquisition site

obtain the change of community residents' mentality

Source

historical data sorting

historical data sorting

interviews with relevant personnel

network

The data collected by the web crawler program written in Python is preprocessed. The preprocessing work is mainly composed of two parts. (1) The text is de duplicated, the mechanical compression is de worded, and the short sentence is deleted to remove the unhelpful sentences and words; (2) The data is cleaned, integrated, transformed and regulated, and then the simulation analysis data required by this topic is constructed. The specific steps are shown in Figure 2.

Figure 2 Network big data crawling and processing process.

Third, the project team held many internal meetings to analyse, screen, supplement, revise and update the influencing factors and indicators obtained in the steps above. Un-marked tertiary indicators in Table 2 are from the discussion and research of our team.

Fourth, the opinions and suggestions of relevant experts in academia and industry were conducted, and the index system was revised and improved. The work of this part can be seen in my student's dissertation (Master's thesis) [52].

The initial indicators are shown in Table 2.

2. We rearranged the serial numbers of tables and figures. See the manuscript.

3. We added reference NO.52 in Page 24 (Lines 787-788).
